# TRAINING TRANSFORMERS WITH ENFORCED LIPSCHITZ BOUNDS

## ABSTRACT

Neural networks are often highly sensitive to input and weight perturbations. This sensitivity has been linked to pathologies such as vulnerability to adversarial examples, divergent training, and overfitting. To combat these problems, past research has looked at building neural networks entirely from Lipschitz components. However, these techniques have not matured to the point where researchers have trained a modern architecture such as a transformer with a Lipschitz certificate enforced beyond initialization. To explore this gap, we begin by developing and benchmarking novel, computationally-efficient tools for maintaining norm-constrained weight matrices. Applying these tools, we are able to train transformer models with Lipschitz bounds enforced throughout training. We find that optimizer dynamics matter: switching from AdamW to Muon improves standard methods—weight decay and spectral normalization—allowing models to reach equal performance with a lower Lipschitz bound. Inspired by Muon's update having a fixed spectral norm, we co-design a weight constraint method that improves the Lipschitz vs. performance tradeoff on MLPs and 2M parameter transformers. Our $\leq 2$-Lipschitz transformer on Shakespeare text reaches validation accuracy 60%. Scaling to 140M parameters, our $\leq 10$-Lipschitz transformer reaches 21% accuracy on internet text. When matching the NanoGPT baseline accuracy of 37.4%, our Lipschitz-bounded network achieves a maximum activation norm of 112, compared to about 1,872 for the unconstrained network. Our Lipschitz transformers train without stability measures, such as layer norm, QK norm, and logit tanh softcapping.

## 1 INTRODUCTION

Lipschitz bounds for neural networks—bounds on the model sensitivity to input perturbations—are of interest for their effect on generalization, robustness (Bartlett et al., 2017; Tsuzuku et al., 2018), and applications such as differential privacy (Béthune et al., 2024). Seminal work (Arjovsky et al., 2016; Cisse et al., 2017; Yoshida & Miyato, 2017; Anil et al., 2019) enforces Lipschitz bounds beyond initialization for MLPs, RNNs, and GANs, but for transformers, the closest work, LipsFormer (Qi et al., 2023), does not constrain weight matrices during training. Without such constraints, large-scale transformer training can become unstable due to attention and output logits growing too large (Wortsman et al., 2024; Dehghani et al., 2023). Numerical instability is typically bandaged over by methods such as QK norm (Henry et al., 2020; Dehghani et al., 2023) or the recent MuonClip optimizer for 1T parameter scale training (Moonshot AI, 2025). We hypothesize that enforcing a Lipschitz bound may directly prevent training instability. In this paper, we ask:

*Can transformers with small, enforced Lipschitz bounds perform well?*
*How does the weight constraint method affect the Lipschitz versus performance tradeoff?*

Enforcing a Lipschitz bound on a transformer is challenging because some components, such as self-attention, are not globally Lipschitz (Kim et al., 2021). We build on Large et al. (2024), which, similar to LipsFormer, enables Lipschitz continuity by reparameterizing residual connections and modifying self-attention; however, the full story is elusive. LipsFormer goes further than Large et al. (2024) by removing layer norm (Ba et al., 2016), but its implementation may make Lipschitz bounds impossible by setting $\epsilon = 0$ in QK norm. In contrast, we remove activation normalization to explore whether training can proceed with no stability measures. Most importantly, we constrain the weights throughout training so that the model's sensitivity remains within a specified Lipschitz bound.

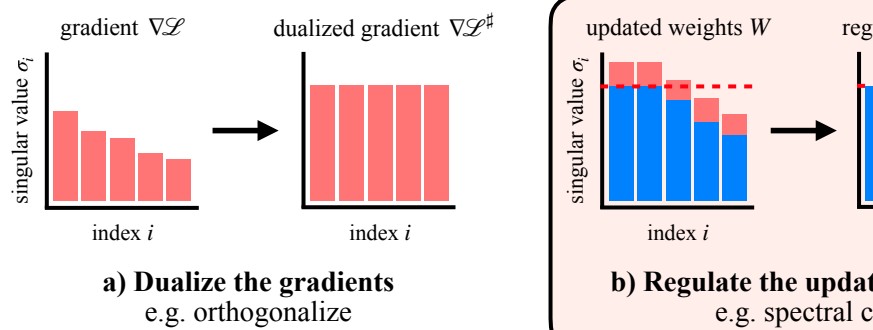

Figure 1: **To train fast and stably, regulate the gradients and also the weights.** Current research considers efficient spectral regulation of gradients, e.g. Muon (Jordan et al., 2024b). A similar opportunity exists for weights. While weight decay with parameter $\lambda$ contracts singular values by a fixed factor $U\Sigma V^\top \mapsto U(1-\lambda)\Sigma V^\top$, more general maps are possible: $U\Sigma V^\top \mapsto Uf(\Sigma)V^\top$ for any nonnegative elementwise function $f$. In this paper, we introduce *spectral cap* to efficiently clip singular values above a threshold $\sigma_{\max} > 0$.

To develop a toolkit for training transformers with an enforced Lipschitz bound, Section 3 compares methods for constraining weight norms. Surprisingly, we find that optimizer choice matters: weight decay (Krogh & Hertz, 1991) and spectral normalization (Yoshida & Miyato, 2017) achieve a better Lipschitz versus performance tradeoff with Muon (Jordan et al., 2024b) than with AdamW (Loshchilov & Hutter, 2019). We observe this effect in MLPs trained on CIFAR-10 (Krizhevsky, 2009) and corroborate it in 2M parameter transformers trained on Shakespeare text (Karpathy, 2022).

Beyond standard weight constraint methods, we are inspired by the Muon optimizer—whose weight updates have small, known spectral norm—to design *spectral soft cap*. This method enforces a desired maximum spectral norm $\sigma_{\max}$ by approximating the map $\sigma \mapsto \min(\sigma_{\max}, \sigma)$ on all singular values $\sigma$ in parallel by iterating odd polynomials on the weights. Theorem B.1 proves that spectrally capping singular values bounds weight norms when training with Muon; we provide no guarantee for AdamW because its update norm varies. For AdamW, we explore a second technique that may be better suited to low stable rank updates. At every step, this technique finds the largest weight singular value and sets it to $\sigma_{\max}$. In analogy with a hammer that strikes the nail that sticks out the most, we call it *spectral hammer*. Our experiments suggest that the best combination is Muon with spectral normalization or spectral soft cap, while for Adam the only competitive technique is spectral hammer.

In Section 4, we begin by training a small transformer on Tiny Shakespeare (Karpathy, 2022); our $\leq$6-Lipschitz model reaches validation loss 1.20. We are not aware of transformers in the literature that have been trained with a Lipschitz bound as low as 6; moreover, we reach loss 1.20 in fewer steps than a recent baseline (Godavarti, 2025). Scaling up to the NanoGPT speedrun benchmark (Jordan et al., 2024a), we train 140M-parameter transformers to competitive performance without layer norm or QK norm. We train a $\leq$10-Lipschitz transformer to 21.2% validation accuracy, compared to 37.4% for a non-Lipschitz baseline. However, to reach a competitive accuracy of 37.4% requires an astronomical global Lipschitz upper bound of $10^{122}$. While Fazlyab et al. (2019) propose tighter Lipschitz bound estimates, inspecting the maximum activation norms reveals that the model operates far from the worst case. On a batch of 393K tokens, the non-Lipschitz baseline has maximum activation entry 1,872 while the $\leq 10^{122}$-Lipschitz model has maximum activation entry 112. Empirically small activations might allow Lipschitz-constrained transformers to enable low-precision training and inference.

In summary, **our contributions are as follows:**

- We train transformers with enforced Lipschitz constraints up to 140M parameters, showing the feasibility of full weight matrix constraints. A $\leq$10-Lipschitz transformer achieves 21% accuracy on FineWeb10B, and a $\leq$2-Lipschitz transformer achieves 60% on Shakespeare.

- We present evidence that weight decay and spectral normalization are more effective with Muon than AdamW, matching accuracy under smaller Lipschitz bounds. A standard robustness experiment holds when using Muon.

- We introduce two weight norm constraint techniques: *spectral capping* and *spectral hammer*. For AdamW, spectral hammer elicits the strongest Lipschitz-constrained performance. For Muon, we prove spectral soft cap bounds weight norm and find it performs similarly or slightly better than spectral normalization.

## 2 RELATED WORK

There is a large literature on the many potential and realized benefits of Lipschitz neural networks (Béthune et al., 2022; Béthune, 2024; Rosca et al., 2020). Input–output Lipschitz certificates help in deployment scenarios where there is a benefit to having strong robustness to input perturbations. Examples include robotic control (O'Connell et al., 2022; Wang et al., 2020), classification in the presence of adversarial input perturbations (Szegedy et al., 2014), and in protocols for AI safety (Brown-Cohen et al., 2024). Input–output Lipschitz certificates are also used in certain generalization guarantees for deep networks (Bartlett et al., 2017; Neyshabur et al., 2018; Dherin et al., 2022).

Similarly, weight–output Lipschitz certificates may be useful in situations where there is an interest in perturbing the weights of a neural network without incurring unstable output behavior. A prime example is stable (Qi et al., 2023; Flynn, 2017) and scalable (Large et al., 2024) training. The recent MuonClip optimizer (Moonshot AI, 2025) similarly addresses exploding attention logits by constraining query and key weights directly; weight norm constraints are the subject of Section 3. A second example is for the design of differentially private training algorithms where there is a need to add carefully calibrated noise to the network weights (Béthune, 2024; Béthune et al., 2024). And a third example is to aid in weight quantization (Elthakeb et al., 2020; Weng et al., 2020).

In fact, there is a close theoretical link between input–output Lipschitzness and weight–output Lipschitzness in deep learning (Large et al., 2024; Béthune, 2024). The reason is that when we compose two subnetworks, Lipschitzness with respect to the weights of the first subnetwork depends upon the degree of input–output Lipschitzness of the second subnetwork.

Various techniques have been proposed for producing neural networks amenable to Lipschitz certification. These include techniques for modifying the architecture and training to improve the resulting Lipschitz properties. For example, spectral normalization (Miyato et al., 2018; Gogianu et al., 2021) has been proposed as a means to control the Lipschitz properties of individual weight matrices. New nonlinearities (Anil et al., 2019) and normalization layers (Qi et al., 2023) have also been proposed.

Furthermore, given a trained model of a given architecture, various techniques have been proposed for deriving Lipschitz certificates. Deriving the exact Lipschitz constant (i.e. the least upper bound) is known to be computationally hard (Katz et al., 2017; Virmaux & Scaman, 2018; Weng et al., 2018) so researchers settle for producing slacker upper bounds. One approach to producing upper bounds—used in this paper for simplicity—is to obtain Lipschitz statements for each component in the architecture and add or multiply them as appropriate to compose them (Szegedy et al., 2014). However, tighter approaches have also been proposed: Weng et al. (2018; 2019) provide examples.

## 3 WEIGHT NORM CONSTRAINTS TO ENFORCE LIPSCHITZ CONSTRAINTS

A function $f(x)$ has Lipschitz bound $K$ under a norm $\|\cdot\|$ if $\|f(x_1) - f(x_2)\| \leq K \cdot \|x_1 - x_2\|$ for all inputs $x_1, x_2$, with the Lipschitz constant being the smallest such $K$. For neural networks, the most common operation is matrix multiplication, which has $\ell_2$ Lipschitz constant equal to the spectral norm of the weight matrix. Prior work has constrained spectral norms through weight decay, spectral normalization, and orthogonal constraints (Krogh & Hertz, 1991; Yoshida & Miyato, 2017; Miyato et al., 2018; Gouk et al., 2021; Jianlin, 2024). These methods, tested with AdamW, have shown benefits for generalization and adversarial robustness (Bartlett et al., 2017; Tsuzuku et al., 2018). The Muon optimizer introduces new possibilities by ensuring small, fixed-norm updates. Inspired by this property, we revisit existing methods and develop new ones for constraining weights. We ask:

> *What is the best way to enforce weight norm constraints throughout training?*

We compare seven methods by their ability to 1) maintain high performance, 2) enforce weight norm constraints, and 3) balance performance with a Lipschitz bound. Overall, we find that Muon achieves

lower Lipschitz bounds and better performance than AdamW. Among constraint methods, spectral soft cap, spectral hard cap, and spectral normalization meet these criteria best.

**Muon enables hard weight constraints.** Unlike AdamW, Muon bounds the weight update norm by the learning rate—if its orthogonalizing polynomial never exceeds 1. We follow (You, 2025) to ensure this property in our experiments. Pethick et al. (2025) noted that bounding the update spectral norm transforms weight decay with parameter $\lambda$ into a *strict* spectral norm constraint of $1/\lambda$ because equilibrium occurs between the update step and weight decay when the weight norm $w$ satisfies $w = w(1 - \lambda\eta) + \eta$ for learning rate $\eta > 0$ (see Section B). We hypothesize this property may explain why Muon improves the Lipschitz vs. performance tradeoff over AdamW for standard methods such as weight decay.

**A spectral generalization of weight decay.** Weight decay can be viewed as a special case of an *odd polynomial iteration* applied to the weights. Odd polynomials are special because they act directly on the singular values: $p(U\Sigma V^\top) = Up(\Sigma)V^\top$, where $U\Sigma V^\top$ is a singular value decomposition. The odd polynomial for weight decay is $p(x) = (1 - \eta\lambda)x$, with learning rate $\eta$ and decay $\lambda$. Cisse et al. (2017) explored an orthogonalizing polynomial $p(x) = (1 + \beta)x - \beta x^3$, but Miyato et al. (2018) note that pressuring all singular values toward one limits spectral information. Their method, *spectral normalization*, enforces norm constraints while allowing singular values below 1 (Gouk et al., 2021), but scales down the entire spectrum. This motivates a more targeted approach: penalizing only the singular values that are too large, leaving smaller ones untouched. Ideally, this applies $\min(\sigma_{\max}, \sigma)$ to each value for maximum norm $\sigma_{\max} \geq 0$, but exact SVD is slow. Odd polynomial iterations offer a fast and effective approximation. We contribute a family of such approximations, *spectral soft cap*, which include weight decay as a special case (derivation in Section B).

### 3.1 METHODS FOR CONTROLLING WEIGHT NORM

The RMS norm is a dimension-independent rescaling of the Euclidean norm, $\|x\|_{\text{RMS}}^2 = \frac{1}{d}\|x\|_2^2$ for $x \in \mathbb{R}^d$, that measures all-ones vectors like $(\pm1, \ldots, \pm1)$ as norm 1. We are interested in controlling the RMS $\to$ RMS operator norm, a rescaled spectral norm shown to be natural in deep learning (Yang et al., 2024; Bernstein & Newhouse, 2025). For a matrix $W \in \mathbb{R}^{d_{\text{out}} \times d_{\text{in}}}$, unit RMS $\to$ RMS norm corresponds to spectral norm of $\sqrt{d_{\text{out}}/d_{\text{in}}}$. While related work uses $\ell_2$ norms, here we report input–output Lipschitz bounds under the RMS $\to$ RMS norm, which convert to $\ell_2$ by multiplying with $\sqrt{d_{\text{output}}/d_{\text{input}}}$ for a given network. We denote the principal singular vector subspace with singular value $\sigma_1 \geq 0$ by $\sigma_1 u_1 v_1^\top$, computed via power iteration. We review existing weight norm constraints and introduce two new methods: spectral capping and spectral hammer.

**Weight decay**, or Frobenius norm regularization, maps $W \mapsto (1 - \lambda\eta)W$ where $\lambda > 0$ is the decay parameter and $\eta > 0$ is the learning rate, guaranteeing a norm bound in conjunction with Muon.

**Spectral weight decay**, or spectral norm regularization, targets only the top singular value, mapping $W \mapsto W - \lambda\sigma_1 u_1 v_1^\top$ where $\lambda > 0$ is the decay parameter (Yoshida & Miyato, 2017; Jianlin, 2024).

**Spectral normalization**, introduced for GAN training (Miyato et al., 2018) and recently applied to LLMs (Zhai et al., 2023; Jha & Reagen, 2024), rescales the weight by its spectral norm during the forward pass. The original implementation leaves the weight unconstrained, so effective update sizes decrease as the weight norm grows. In this work, we instead apply the mapping $W \mapsto (\sigma_{\max}/\max(\sigma_1, \sigma_{\max}))W$, ensuring weights have spectral norm at most $\sigma_{\max}$, but possibly less.

**Stiefel manifold projection** pressures all singular values toward 1 using an odd polynomial iteration $W \mapsto p(W)$. We follow You (2025), whose polynomial converges faster than the polynomial from (Cisse et al., 2017). Although Stiefel manifold projections are typically defined for rectangular matrices, we use the term here for both square and rectangular matrices.

We extend these ideas with two new methods:

**Spectral hammer** is similar to spectral weight decay but sets the top singular value to $\sigma_{\max}$ by mapping $W \mapsto W + (\sigma_{\max} - \sigma_1)u_1 v_1^\top$, like a hammer striking the nail that sticks out most. It does not guarantee the spectral norm stays below $\sigma_{\max}$ because multiple singular vectors may increase per update. Spectral hammer is better suited to low stable rank updates, common in Adam (Zhao et al., 2024), whereas Muon updates are always high stable rank.

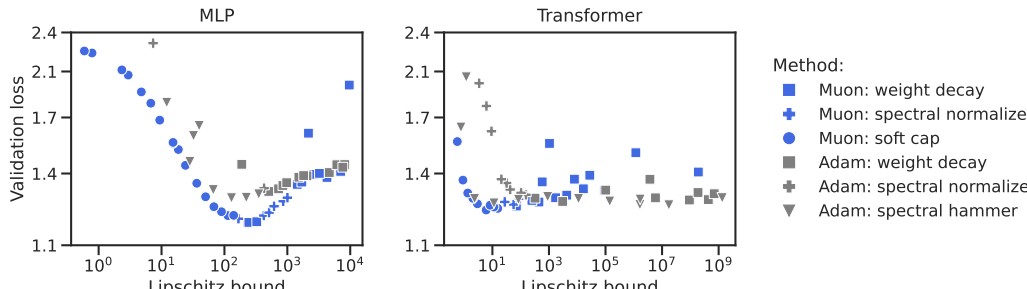

Figure 2: **Muon improves the Lipschitz vs. performance tradeoff for standard weight regularization.** We train 3385 MLPs on CIFAR-10 (left) and 800 transformers on Shakespeare text (right), varying the optimizer and weight constraints. Weight decay and spectral normalization reach lower loss at smaller Lipschitz bounds with Muon. Our new methods—spectral soft cap and spectral hammer—also show promise. See Section F for experimental details.

**Spectral capping** is designed for Muon's high stable rank updates, approximating $\sigma \mapsto \min(\sigma_{\max}, \sigma)$ for all singular values in parallel with odd polynomials instead of costly SVDs. Our main variant, *spectral soft cap*, applies a loose approximation $p_2(p_1(x))$, where $p_1(x) = x - \alpha x^3$ and $p_2(x) = x + \alpha x^3$ with strength parameter $\alpha \geq 0$, as discussed in Section B. Weight decay can be used as a preliminary step by applying $p_0(x) = (1 - \lambda\eta)x$. This composition is designed to decay a singular value very little when $\sigma \ll \sigma_{\max}$, but when $\sigma = \sigma_{\max}$, to decay it enough to counteract Muon's update norm, strictly enforcing the bound. With scheduled learning rates, finding the minimal $\alpha \geq 0$ prevents error accumulation.

**Theorem 3.1** (Spectral soft cap bounds spectral norm when training with Muon). *Suppose we wish to bound the spectral norm of a weight matrix $W$ to never exceed $\sigma_{max} > 0$. Let $\eta > 0$ be the learning rate and $\lambda > 0$ be the weight decay. If $\|W\|_* \leq \sigma_{max}$, then there exists a minimal $\alpha_* \geq 0$ such that performing the following three operations ("the training step") preserves $\|W\|_* \leq \sigma_{max}$:*

1. *Weight decay: $W \mapsto W \cdot (1 - \lambda\eta)$.*
2. *Muon update: $W \mapsto W + \Delta W$, where $\|\Delta W\|_* \leq \eta$.*
3. *Spectral soft cap: $W \mapsto p_2(p_1(W))$, where $p_1(x) = x - \alpha_* x^3$ and $p_2(x) = x + \alpha_* x^3$.*

*Calculating $\alpha_*$ reduces to solving for the roots of a quartic polynomial.*

Section B gives the proof. Training using the minimal strength parameter $\alpha_* > 0$ indefinitely bounds weight matrix spectral norm. In practice, Muon scales the learning rate by $\sqrt{d_{\text{out}}/d_{\text{in}}}$, so the theorem instead bounds the $\text{RMS} \to \text{RMS}$ operator norm of the weight matrix.

A second variant, *spectral hard cap*, uses the matrix sign function to approximate $\sigma \to \min(\sigma_{\max}, \sigma)$ on the singular values, as discussed in Section C. Because this approximation is fixed, errors can compound late in training as the learning rate decays to 0. Since spectral soft cap and spectral hard cap are designed for Muon, we did not test them with AdamW.

Together, these methods cover a range of trade-offs between strict norm enforcement, preserving the spectrum, and computational efficiency. Other approaches may prove better, making this an interesting direction for future work.

### 3.2 ADAMW AND MUON: COMPARING WEIGHT CONSTRAINT METHODS

In Figure 2, we show the tradeoff between validation loss and Lipschitz bound, calculated with respect to the $\text{RMS} \to \text{RMS}$ operator norm (Section 3.1). For ReLU MLPs, the Lipschitz bound is the product of the $\text{RMS} \to \text{RMS}$ weight norms. For a transformer, the Lipschitz bound is calculated as described in Section 4.2. Muon consistently achieves both lower validation loss *and* lower Lipschitz bounds than AdamW on CIFAR-10 MLPs and Shakespeare transformers, motivating our choice to adopt Muon for larger-scale experiments. Spectral normalization and spectral soft cap make the most efficient use of a Lipschitz budget. Spectral hammer, which is best for AdamW's low stable rank weight updates, performs competitively but does not enforce a bound, limiting its reliability where

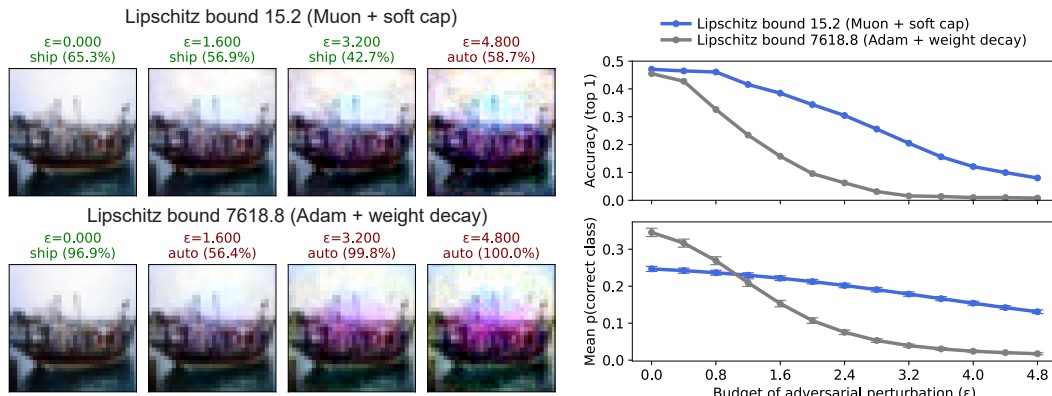

Figure 3: **Networks trained with Muon and spectral soft cap have lower Lipschitz bounds and greater adversarial robustness.** Lower Lipschitz bounds have been linked to robustness (Cisse et al., 2017; Huang et al., 2021). We train a CIFAR-10 MLP with Lipschitz bound 15.2 (Muon + spectral soft cap), matching the 45% clean accuracy of a baseline model (AdamW + weight decay) with a much higher Lipschitz bound of 7618.8. Left: Adversarial attacks with varying $\ell_2$ budget $\epsilon$. Top right: Robustness across 2000 test images measured by top-1 accuracy against $\epsilon$. The Lipschitz-constrained model maintains a higher accuracy for larger values of $\epsilon$. Bottom right: The mean correct class probability in the Lipschitz-constrained network starts below the baseline model's but degrades slowly under increasing $\epsilon$. By contrast, the baseline model peaks at $\epsilon = 0$ and drops off sharply.

constraint enforcement is critical. The sweep includes 4185 runs, reporting the best validation loss per Lipschitz bin.

## 3.3 ADVERSARIAL ROBUSTNESS OF LIPSCHITZ NETWORKS

Prior work (Cisse et al., 2017; Huang et al., 2021) links adversarial robustness to a network's Lipschitz constant, suggesting Lipschitz control as a potential path to high certified accuracy. We confirm this relationship holds for MLPs trained with Muon and spectral soft cap. Figure 3 compares two MLPs with similar baseline validation accuracy ($\approx 45\%$) but different Lipschitz bounds: 15.2 (Muon + spectral soft cap) vs. 7618.8 (AdamW + weight decay).

Both models achieve similar clean accuracy, but the Lipschitz-constrained MLP shows a smoother drop in accuracy and confidence on the first 2000 CIFAR-10 test images as adversarial $\ell_2$ perturbations increase. Larger $\epsilon$ values are required to fool the constrained network (Figure 3, left).

## 3.4 COMPARING WEIGHT CONSTRAINT METHODS WITHIN MUON

In Figure 4, we use Muon alongside all seven weight constraint methods to test three goals: 1) defining a Lipschitz bound before training, 2) enforcing that bound throughout training, and 3) matching or exceeding standard weight decay performance. Each method is evaluated on a 3-layer, 256-hidden dimension MLP trained with Muon on CIFAR-10. Full experimental details are in Section F.

In Figure 4 (left), spectral normalization, spectral soft cap, and spectral hard cap form the Lipschitz vs. validation loss frontier. In Figure 4 (middle), we see the best hyperparameter settings of spectral normalization, spectral soft cap, and spectral hard cap are the only methods to reach within 1% accuracy of the baseline (Muon with weight decay); others fall within 4%.

Figure 4 (right) shows the evolution of hidden layer norms during training. All methods except spectral hammer keep norms at or under their target values. Spectral hammer exceeds its target but trends downward late in training, indicating it may work under Muon with longer runs and scheduled learning rates. Within our 50-epoch window, however, it fails to reliably control the Lipschitz bound. Spectral weight decay does not enforce predefined Lipschitz bounds and is omitted.

We select soft cap, hard cap, and spectral normalization for transformer experiments. To isolate their effects, we do not combine them with weight decay, though such combinations may help.

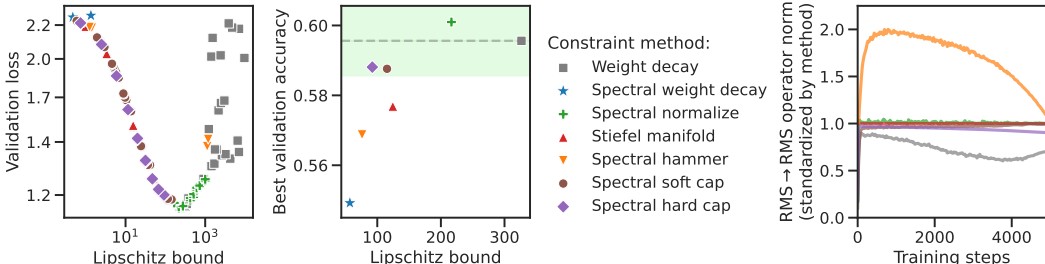

Figure 4: **Left: Weight constraint methods designed for Muon lie on the Lipschitz vs. loss frontier.** Each point shows the lowest validation loss achieved at a given bound for CIFAR-10 MLPs. In the low-loss regime, staying on the frontier (left half of the parabola) requires spectral normalization or spectral capping. **Middle: Spectral normalization and spectral capping match baseline accuracy at lower bounds.** Each method falls within 1% accuracy (shaded green region) while reducing the bound. **Right: RMS → RMS norms of hidden layers during training.** Norms are standardized to target 1. Spectral normalization and Stiefel manifold projection hit this target exactly; weight decay, soft cap, and hard cap stay below the target; spectral hammer fails to constrain.

## 4 TRANSFORMERS WITH ENFORCED WEIGHT CONSTRAINTS

To develop a toolkit for training Lipschitz-constrained transformers, we first address residual connections and self-attention (Section 4.1), then describe how to calculate a Lipschitz bound for a transformer (Section 4.2). In Section 4.3, spectral normalization and spectral soft cap perform well on 2M-parameter Shakespeare transformers. In Section 4.4, we scale up to 140M-parameter transformers trained on FineWeb10B (Penedo et al., 2024), starting from the competitive NanoGPT speedrun benchmark (Jordan et al., 2024a) to avoid undertuned baselines.

### 4.1 BREAKING THE MULTIPLICATION BARRIER?

A major triumph of Large et al. (2024) is transformers with depth-independent Lipschitz bounds. However, their approach assumes activations with unit RMS norm. Since we later relax this constraint, our transformers will generally *not* have a depth-independent Lipschitz bound. Nonetheless, we use their two architectural modifications.

**Reparameterizing residual connections.** The classic residual connection from He et al. (2016) defines the update $x + \text{block}(x)$, which can exponentially inflate the Lipschitz bound: even $x + \text{identity}(x)$ doubles it every layer. Large et al. (2024) avoid this by using the convex combination

$$\frac{N-1}{N} \cdot x + \frac{1}{N} \cdot \text{block}(x) \tag{1}$$

where $N$ is the number of layers, which is 1-Lipschitz if the block is 1-Lipschitz (Proposition 4 of Large et al. (2024)). We adopt this parameterization, but the bound fails when activation norms exceed 1. Since strict 1-Lipschitz constraints hurt performance, we do not fully break the multiplication barrier: deeper networks can accrue astronomical Lipschitz bounds.

**Attention with $1/d$ scaling.** Original multihead attention from Vaswani et al. (2017) has no global Lipschitz bound (Kim et al., 2021). Vaswani et al. (2017) used $1/\sqrt{d}$ scaling because two random vectors $u, v$ with mean 0 and variance 1 will have a dot product $u \cdot v$ with mean 0 and variance the dimension $d$. But key-query pairs may align more strongly. Perfect alignment suggests $1/d$ scaling. Large et al. (2024) prove that

$$\text{softmax}\left(\frac{QK^\top}{d}\right) V, \tag{2}$$

is 1-Lipschitz under unit input norms, with respect to $\|\cdot\|_{\infty\text{RMS}}$, the max RMS norm of any token. Multiplying by $\frac{1}{3}$ makes attention composed with the 3-sensitive input tuple $(q, k, v)$ unit sensitivity (Proposition 7 of Large et al. (2024)). Unlike Large et al. (2024), we remove layer norm so that every operation is Lipschitz continuous.

## 4.2 Calculating the Lipschitz bound of a transformer

To test whether small-Lipschitz transformers can perform well, we would need to know what weight norm yields a desired Lipschitz bound. This section sketches an algorithm for bounding the transformer's Lipschitz constant given its weight norms; details are in Section D. Our bound tightens Theorem 2 from LipsFormer (Qi et al., 2023) by using individual weight norms rather than the maximum across residual blocks. Fazlyab et al. (2019) suggest ways to further tighten such bounds, which we leave to future work. To bound self-attention, we extend Proposition 7 of (Large et al., 2024) to inputs that are not unit norm, an essential concession to explore Lipschitz vs. performance tradeoffs.

Our Lipschitz bounds are with respect to the max RMS norm over token positions, denoted $\|\cdot\|_{\infty\text{RMS}}$.

**Step 1: Bound activation norms.** Our Lipschitz bound for rescaled dot-product attention relies on the activation norms staying finite. We compute a per-layer bound on the max $\|\cdot\|_{\infty\text{RMS}}$ activation norm by combining residual connections with per-block max activation increases: for an MLP by multiplying its two weight norms and for attention by multiplying its $W_V$ and $W_O$ norms. Our MLPs slightly decay activation norms by scaling GeLU by its maximum derivative, GeLU/1.1289.

**Step 2: Bound the Lipschitz constant.** Let $L$ be the Lipschitz constant before a layer. After a residual connection, it is at most $(1-\alpha)L + \alpha \cdot L \cdot L_{\text{block}}$. For MLPs we calculate $L_{\text{block}} = \|W_{\text{in}}\|_{\text{RMS}\to\text{RMS}} \cdot \|W_{\text{out}}\|_{\text{RMS}\to\text{RMS}}/1.1289$ from the rescaled GeLU. For attention, we use Theorem D.1.

## 4.3 Shakespeare Transformer

Before scaling to NanoGPT, we explore the Lipschitz vs. performance tradeoff in smaller transformers, aiming for models with small Lipschitz bounds *at every layer*. While Béthune et al. (2022) comment that any $L$-Lipschitz classifier can be made 1-Lipschitz by dividing the logits by $L$, we do not rescale logits, although adjusting scaling temperature during training could have beneficial effects (Agarwala et al., 2023). Full experimental details are in Section F.

**We can train a competitive $\leq$2-Lipschitz Shakespeare transformer**. Our $\leq$2-Lipschitz transformer reaches a validation loss of 1.29, compared to 1.47 for the Karpathy (2022) baseline, though the baseline may be undertuned. Our model has dimension 256, depth 3, and was trained for 2000 steps with Muon; the baseline has dimension 384, depth 6, and was trained for 5000 steps with AdamW. Our transformer does not use layer normalization. Achieving this performance requires relaxing $\sigma_{\max}$ to around 2. The best validation loss in our sweep was 1.20 with a $\leq$6.02-Lipschitz transformer, surpassing any baseline we are aware of, such as the 1.23 loss of Godavarti (2025) while training on 300x fewer tokens. Some gains may reflect hyperparameters or optimizer choice, but this demonstrates a performant transformer with a small enforced Lipschitz bound.

## 4.4 Scaling to NanoGPT

We validate our methods by training a 140M-parameter transformer on top of the NanoGPT speedrun benchmark (Jordan et al., 2024a). The baseline is tuned to reach validation loss 3.28 in the shortest wallclock time. As of February 1, 2025, the record uses 0.7B tokens, or 3 minutes of training on an 8xH100, and achieves 39.4% accuracy. We implement our methods on top of the speedrun while keeping all other training methods fixed. We compare against 1) the original speedrun, 2) the "NanoGPT" baseline where we remove speedrun-specific architectural decisions such as skip connections and learnable scale parameters, and 3) the "Modula" baseline with residual reparameterization and $\frac{1}{d}$ attention scaling from Section 4.1. We also replace ReLU$^2$ activations with GeLU/1.1289—making the activation function Lipschitz continuous. We report validation loss and accuracy as well as the max activation norm as primary comparison metrics.

To implement our method, we remove layer norms, logit tanh softcaps, and QK norms from the Modula baseline and implement our weight constraints on top. At initialization, we project linear weights to be semi-orthogonal and normalize the embeddings to RMS norm 1. Finally, unlike LipsFormer (Qi et al., 2023), we enforce weight norm constraints throughout training: embeddings are capped to norm 1, and all other weights are constrained after each step using the methods in Section 3.1. Linear layers in the MLPs and attention use fp8 precision *without the need for tensorwise or blockwise scaling*.

| Transformer Architecture | Lipschitz Bound | Tokens Used | Weight Constraint | Validation Accuracy ($\uparrow$) | Validation Loss ($\downarrow$) | Activation Max Entry | Activation Max RMS |
|---|---|---|---|---|---|---|---|
| Baseline (NanoGPT) | $\infty$ | 0.7B | none | **0.380** | **3.410** | 91648 | 24092 |
| Baseline (Modula) | $\infty$ | 0.7B | none | *0.374* | 3.491 | 1872 | 110.3 |
| Ours ($\sigma_{max} = \infty$) | $\infty$ | 0.7B | none | nan | nan | nan | nan |
| LipsFormer | $10^{130}$ | 0.7B | none | 0.301 | 4.130 | 61 | - |
| Ours ($\sigma_{max} = 8$) | $10^{134}$ | 0.7B | spectral normalize | 0.362 | 3.582 | 37.75 | 6.1 |
| Ours ($\sigma_{max} = 8$) | $10^{122}$ | 1.4B | spectral soft cap | *0.374* | *3.481* | 112 | 28 |
| Ours ($\sigma_{max} = 1$) | 10 | 0.7B | spectral normalize | 0.212 | 5.047 | **6.5** | **1** |
| Baseline (Speedrun) | $\infty$ | 0.7B | none | 0.394 | **3.280** | 148480 | 12480 |
| Ours ($\sigma_{max} = 16$) | $10^{264}$ | 2.8B | spectral normalize | *0.395* | *3.280* | **49.3** | **7.1** |

Table 1: **Transformers with enforced Lipschitz constraints can match NanoGPT performance.** The NanoGPT speedrun is a competitively tuned GPT-2 replication (Karpathy, 2022; Jordan et al., 2024a). Using it as a baseline, we substitute Lipschitz transformer components and constrain weight norms by $\sigma_{max} \geq 0$. Unlike LipsFormer (Qi et al., 2023), our constraints enforce a bound chosen prior to training. A $\leq 10$-Lipschitz transformer trains stably to 21.2% accuracy without layer norm, QK norm, or tanh on logits. The same model without weight constraints diverges. Matching baseline accuracy, however, requires a Lipschitz bound of $10^{122}$, computed as in Section 4.2. Our Lipschitz bounds may be loose, as suggested by the small maximum activation we observe across a batch of 393K tokens. Final loss variance is 0.0008.

Table 1 summarizes our 140M-parameter results. Unsurprisingly, training diverges without weight constraints. Unlike LipsFormer, our method enforces a Lipschitz bound specified before training. The key lever is the maximum $\text{RMS} \to \text{RMS}$ norm $\sigma_{max}$ for linear layers. Smaller $\sigma_{max}$ values correspond to tighter Lipschitz bounds but may reduce performance. The attention and final logit scale also affect the bound. With spectral normalization, $\sigma_{max} = 1$ and final logit scale 8, we train a $\leq 10$-Lipschitz transformer to validation loss 5.047 and accuracy 21.2%. No activation in this model exceeds RMS norm 1, aiding stability during training. Raising $\sigma_{max}$ to 8 and using $2\times$ training data matches Modula baseline performance while keeping all activation entries comfortably within the fp8 range. Further raising $\sigma_{max}$ to 16 and using $4\times$ training data matches speedrun validation performance while achieving max activation entry 49.3, compared to about 148K. The NanoGPT baseline has a max activation entry of about 92K, demonstrating higher sensitivity than our methods.

## 5 DISCUSSION

Despite large Lipschitz bounds, our NanoGPT transformers exhibit low maximum activation entries (6.5–112) compared to the baseline (1,872). This may explain their stable training free from standard measures including layer norm, QK norm, or tanh logit softcapping. Constrained network activation entries never exceed the representable range of FP4 E3M0 format. Future work can test whether these low activations enable low-precision training and whether stability persists at larger scales.

For MLPs and small transformers, we find that using Muon improves the Lipschitz vs. performance tradeoff. Out of the weight constraint methods we test, *spectral normalization*, *spectral soft cap*, and *spectral hard cap* compare favorably to standard weight decay. Perhaps surprisingly, on both CIFAR-10 and Shakespeare data, we achieve our best loss with Lipschitz-enforced models, potentially representing a training speed benefit.

Our work has several limitations. We did not find a principled way to select weight norm, final logit scale, and attention logit scale hyperparameters, instead relying on sweeps. Our Lipschitz bound also increases rapidly as depth increases, unless we constrain weights to unit norm. A different architecture, or insight beyond a global Lipschitz bound, could make progress on this problem.

In conclusion, this paper develops a method for training transformers with an enforced Lipschitz bound throughout training, extending earlier efforts focused on different architectures or only constraining at initialization. Lipschitz-certified transformers may be of particular interest for domains such as privacy, control, adversarial robustness, low-precision training, and loss-spike-free, large-scale pretraining. Although training speed benefits fade in our NanoGPT speedrun experiments, we wonder whether at this scale Lipschitz-enforced training can also be made faster than standard training.

## 6 AUTHOR STATEMENTS

### 6.1 ETHICS STATEMENT

The authors have read the ICLR code of ethics and declare that we conform to them. We have not identified any direct major ethical risks of this project.

### 6.2 REPRODUCIBILITY STATEMENT

We provide multiple resources to reproduce our experiments. In Section A, we provide a link to an anonymized GitHub repository that includes code for reproducing all of our results. This includes files for training Lipschitz-constrained MLPs and transformers using our methods and implementations, as well as code for data loaders that we use to access the public datasets we used for experiments in this paper. In Section F, we explain the hyperparameters that we used for our coordinate descent hyperparameter search. We also cite our datasets and explain the compute resources we used for our experiments. Our two main theoretical contributions have full proofs in Section B and Section C. We describe our method for calculating Lipschitz bounds for transformers in Section D. All together, the supplementary materials allow a reader to reproduce our results.

### 6.3 LLM USAGE STATEMENT

Our ideas, results, and writing are due to humans, not LLMs. We used LLMs to implement some standard methods, such as power iteration. The majority of our code was authored by humans; the rest was tested and verified. LLMs were used to assist in coding the figure generation scripts. After we completed writing the paper, LLMs were sometimes used to make the writing more concise. These suggestions were reviewed by humans and never copied into our document. Overall, our use of LLMs was to automate standard procedures and was orthogonal to the originality and results of our paper.

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

## A   ANONYMOUS CODE LINK

Our code is available on an anonymized GitHub repository: `https://anonymous.4open.science/r/lipschitz-transformers-A7B3`.

# B   COUPLING SPECTRAL CAP TO LEARNING RATE

The intuition for the proof can be summarized, "There is a maximum amount the singular values of a weight can grow after a weight update; as long as the minimum decrease of too-large singular values exceeds their maximum growth, then the set of weight singular values stays bounded forever."

**Theorem B.1** (Spectral soft cap bounds spectral norm when training with Muon). *Suppose we wish to bound the spectral norm of a weight matrix $W$ to never exceed $\sigma_{max} > 0$. Let $\eta > 0$ be the learning rate and $\lambda > 0$ be the weight decay. If $\|W\|_* \leq \sigma_{max}$, then there exists a minimal $\alpha_* \geq 0$ such that performing the following three operations ("the training step") preserves $\|W\|_* \leq \sigma_{max}$:*

1. *Weight decay: $W \mapsto W \cdot (1 - \lambda\eta)$.*

2. *Muon update: $W \mapsto W + \Delta W$, where $\|\Delta W\|_* \leq \eta$.*

3. *Spectral soft cap: $W \mapsto p_2(p_1(W))$, where $p_1(x) = x - \alpha_* x^3$ and $p_2(x) = x + \alpha_* x^3$.*

*Calculating $\alpha_*$ reduces to solving for the roots of a quartic polynomial.*

*Proof.* Let $\sigma_1 \geq \cdots \geq \sigma_n$ be the singular values of a weight matrix $W \in \mathbb{R}^{d_{in} \times d_{out}}$, where $W$ has bounded spectral norm $\|W\|_* \leq \sigma_{max}$. Weyl's inequality in linear algebra says that the singular values of the updated weight $W + \Delta W$ cannot change by more than the spectral norm of $\Delta W \in \mathbb{R}^{d_{in} \times d_{out}}$. Specifically, the singular values of $W + \Delta W$ fall in the range $\sigma_i' \in [\sigma_i - \|\Delta W\|_*, \sigma_i + \|\Delta W\|_*]$.

The weight update in Muon has bounded spectral norm $\|\Delta W\| \leq \eta$, because $\Delta W$ is an orthogonal matrix (the "o" in Muon) scaled by the learning rate $\eta > 0$. Therefore, every singular value of $W$ after adding $\Delta W$ cannot increase by more than $\eta$ ("max increase").

We seek a minimal spectral cap strength parameter $\alpha_* \geq 0$ such that performing the following three operations ("the training step"), collectively referred to as $W \mapsto \Phi(W, \alpha_*)$, preserves $\|W\|_* \leq \sigma_{max}$:

1. Weight decay: $W \mapsto W \cdot (1 - \lambda\eta)$.

2. Muon update: $W \mapsto W + \Delta W$, where $\|\Delta W\|_* \leq \eta$.

3. Spectral soft cap: $W \mapsto p_2(p_1(W))$, where $p_1(x) = x - \alpha_* x^3$ and $p_2(x) = x + \alpha_* x^3$.

The remainder of the proof will show that the "minimum decrease" of singular values due to steps (1) and (3) is a monotonically increasing function in the singular value $\sigma$. In other words, larger singular values are reduced more strongly. Then since step (2) cannot raise singular values by more than a fixed amount $\eta$, performing all three steps cannot raise singular values after a certain point. The final observation will be that this point occurs at the threshold $\sigma_{max} > 0$ when we choose a minimal strength parameter $\alpha_* \geq 0$ accordingly. Thus the training step satisfies $\|\Phi(W, \alpha_*)\|_* \leq \sigma_{max}$ if $\|W\|_* \leq \sigma_{max}$, which will prove that the weight norm remains bounded after each training step.

To begin, we observe that $p(\sigma) = p_2(p_1(\sigma))$ has derivative bounded above like $p'(\sigma) \leq 1$ for $\sigma \geq 0$. Furthermore it has non-positive second derivative $p''(\sigma) \leq 0$ for $\sigma \geq 0$. These two facts together prove that the decrease $\sigma - p(\sigma)$ due to spectral soft cap is a monotonically increasing function of $\sigma$. Thus, after some threshold $\sigma_*$, all singular values $\sigma > \sigma_*$ will decrease after the training step.

All that is left is to calculate the minimal coupling strength $\alpha_*$ that causes this threshold to be $\sigma_* = \sigma_{max}$. To do so, we solve for $\alpha$ in the polynomial equation

$$p(\sigma_{max} \cdot (1 - \lambda\eta) + \eta) = \sigma_{max}.$$

The equation represents performing all three operations in the training step and finding that the singular value does not change. Let us denote the largest that $\sigma_{max}$ can become after steps (1) and (2) by $k = \sigma_{max} \cdot (1 - \lambda\eta) + \eta$. Expanding, the polynomial equation becomes

$$-k^9\alpha_*^4 + 3k^7\alpha_*^3 - 3k^5\alpha_*^2 + k - \sigma_{max} = 0.$$

This is a quartic polynomial in $\alpha$. If $\sigma_{max} < 1/\lambda$, then $k < \sigma_{max}$ means we are done since weight decay already causes the singular value $\sigma_{max}$ to decrease, and step (3) will only decrease it further. Otherwise if $\sigma_{max} \geq 1/\lambda$, the quartic polynomial will have a solution by the mean value theorem,

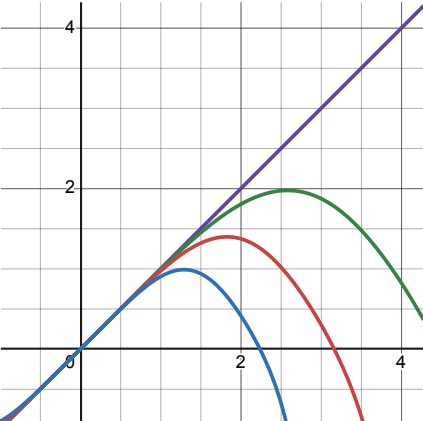

Figure 5: **Spectral soft cap** is a weight constraint method that applies the above odd polynomial to a weight matrix, which applies it to all singular values in parallel. First it applies $p_1(x) = x - \alpha x^3$, then it applies $p_2(x) = x + \alpha x^3$. The composition is depicted for $\alpha = 0.2$ (blue), $\alpha = 0.1$ (red), $\alpha = 0.05$ (green), $\alpha = 0$ (purple).

since it evaluates to a nonnegative number at $\alpha = 0$ but its leading term is negative. Thus there exists a minimal solution called $\alpha_*$ that preserves $\|W\|_* \le \sigma_{\max}$ in every training step, as desired. □

One limitation of automatic coupling is that it may be stronger than necessary, because it assumes updates align perfectly with the weights in the worst case. If the learning rate is scheduled to 0, gradients may align less with the existing weights especially at the end, which can cause the weight norm to contract slightly.

## C    SPECTRAL CLIPPING

Spectral clipping is a generalization of spectral cap that puts an upper *and* lower bound on the singular values. It maps $\sigma \mapsto \text{clip}(\sigma, \sigma_{\min}, \sigma_{\max})$, where $0 < \sigma_{\min} < \sigma_{\max}$ (Cesista, 2025; Su, 2025).

**Definition C.1** (Spectral clipping). Let $W \in \mathbb{R}^{m \times n}$ and $W = U\Sigma V^T$ be its singular value decomposition, where $\Sigma = (\sigma_1, \ldots, \sigma_{\min(m,n)})$ are the singular values of $W$. Then we define spectral clipping as the matrix function $\text{spectral\_clip}_{[\alpha,\beta]} : \mathbb{R}^{m \times n} \to \mathbb{R}^{m \times n}$ acting on the singular values of $W$,

$$\text{spectral\_clip}_{[\alpha,\beta]}(W) = U\text{clip}_{[\alpha,\beta]}(\Sigma)V^T, \tag{3}$$

where $\alpha \leq \beta$ and $\alpha, \beta \in \mathbb{R} \cup \{-\infty, \infty\}$ control the minimum and maximum attainable singular values. The clip function $\text{clip}_{[\alpha,\beta]} : \mathbb{R} \to \mathbb{R}$ is applied element-wise on the singular values of $W$,

$$\text{clip}_{[\alpha,\beta]}(x) = \begin{cases} \alpha & \text{if } x < \alpha \\ x & \text{if } \alpha \leq x \leq \beta \\ \beta & \text{if } \beta < x \end{cases} . \tag{4}$$

We could compute spectral clipping via SVD. However, SVD does not take full advantage of GPU tensor cores and typically requires casting the inputs to full precision, making it slow in practice. Following Cesista (2025), we can instead spectrally clip via the matrix sign function used by Muon (Jordan et al., 2024b) because of the following identity,

$$\text{clip}_{[\alpha,\beta]}(x) = \frac{1}{2}[\alpha + \beta + (\alpha - x)\text{sign}(\alpha - x) - (\beta - x)\text{sign}(\beta - x)]. \tag{5}$$

Thus,

$$\begin{aligned}
\text{spectral\_clip}_{[\alpha,\beta]}(W) &= U\text{clip}_{[\alpha,\beta]}(\Sigma)V^T \\
&= U\frac{1}{2}[(\alpha + \beta)I + (\alpha I - \Sigma)\text{sign}(\alpha I - \Sigma) \\
&\quad - (\beta I - \Sigma)\text{sign}(\beta I - \Sigma)]V^T \\
&= \frac{1}{2}[(\alpha + \beta)UV^T + U(\alpha I - \Sigma)\text{sign}(\alpha I - \Sigma)V^T \\
&\quad - U(\beta I - \Sigma)\text{sign}(\beta I - \Sigma)V^T] \\
&= \frac{1}{2}[(\alpha + \beta)UV^T \\
&\quad + U(\alpha I - \Sigma)(V^T V)\text{sign}(\alpha I - \Sigma)(U^T U)V^T \\
&\quad - U(\beta I - \Sigma)(V^T V)\text{sign}(\beta I - \Sigma)(U^T U)V^T] \\
&= \frac{1}{2}[(\alpha + \beta)UV^T \\
&\quad + (\alpha UV^T - U\Sigma V^T)(V\text{sign}(\alpha I - \Sigma)U^T)(UV^T) \\
&\quad - (\beta UV^T - U\Sigma V^T)(V\text{sign}(\beta I - \Sigma)U^T)(UV^T)] \\
&= \frac{1}{2}[(\alpha + \beta)UV^T \\
&\quad + (\alpha UV^T - U\Sigma V^T)(U\text{sign}(\alpha I - \Sigma)V^T)^T(UV^T) \\
&\quad - (\beta UV^T - U\Sigma V^T)(U\text{sign}(\beta I - \Sigma)V^T)^T(UV^T)] \\
\text{spectral\_clip}_{[\alpha,\beta]}(W) &= \frac{1}{2}[(\alpha + \beta)I \\
&\quad + (\alpha \cdot \text{msign}(W) - W)\text{msign}(\alpha \cdot \text{msign}(W) - W)^T \\
&\quad - (\beta \cdot \text{msign}(W) - W)\text{msign}(\beta \cdot \text{msign}(W) - W)^T \\
&\quad ]\text{msign}(W). \tag{6}
\end{aligned}$$

## C.1 Spectral hardcapping

Spectral hardcapping is a special case of spectral clipping where $\alpha \leq 0$. And since singular values are always nonnegative, setting $\alpha = -\beta$ simplifies Equation (5) to

$$\text{clip}_{[-\beta,\beta]}(x) = \frac{1}{2}[\beta + x - (\beta - x)\text{sign}(\beta - x)]. \tag{7}$$

We can now construct a spectral hardcapping formula in terms of the well-known msign function:

$$\text{spectral\_hardcap}_\beta(W) = U\text{clip}_{[-\beta,\beta]}(\Sigma)V^T$$

$$= U\frac{1}{2}[\beta I + \Sigma - (\beta I - \Sigma)\text{sign}(\beta I - \Sigma)]V^T$$

$$= \frac{1}{2}[\beta UV^T + U\Sigma V^T$$

$$- U(\beta I - \Sigma)(V^T V)\text{sign}(\beta I - \Sigma)(U^T U)V^T]$$

$$= \frac{1}{2}[\beta UV^T + U\Sigma V^T$$

$$- (U(\beta I - \Sigma)V^T)(U\text{sign}(\beta I - \Sigma)V^T)^T(UV^T)]$$

$$= \frac{1}{2}[\beta \text{msign}(W) + W$$

$$- (\beta\text{msign}(W) - W)\text{msign}(\beta\text{msign}(W) - W)^T\text{msign}(W)]$$

$$\text{spectral\_hardcap}_\beta(W) = \frac{1}{2}[\beta\text{msign}(W) + W$$

$$- \text{msign}(\beta I - \text{msign}(W)W^T)(\beta\text{msign}(W) - W)]. \tag{8}$$

The last equality follows from the transpose equivariance and unitary multiplication equivariance of odd analytic matrix functions acting on singular values.

## C.2 Spectrally clipped weight decay

As a further extension, we can use spectral hardcapping to construct a spectrally clipped weight decay. Unlike standard weight decay, spectrally clipped weight decay only applies the decay term to singular values larger than a threshold $\beta$, chosen *a priori*:

$$\text{clipped\_weight\_decay}_{\lambda,\beta}(\Sigma) = \begin{cases} \Sigma & \text{if } \Sigma \leq \beta \\ (1-\lambda)\Sigma + \lambda\beta & \text{if } \Sigma > \beta \end{cases} \tag{9}$$

$$= (1-\lambda)\Sigma + \lambda \cdot \text{clip}_{[0,\beta]}(\Sigma).$$

Thus,

$$\text{spec\_clipped\_weight\_decay}_{\lambda,\beta}(W) = U\text{clipped\_weight\_decay}_{\lambda,\beta}(\Sigma)V^T \tag{10}$$

$$= (1-\lambda)W + \lambda \cdot \text{spectral\_hardcap}_\beta(W) \tag{11}$$

Following the argument made in Section B, we can derive the equilibrium point of spectrally clipped weight decay as follows.

**Proposition C.0.1** (Equilibrium point of spectrally clipped weight decay). *Let $\eta \in (0,\infty)$ be the learning rate, $\lambda \in (0,1]$ be the decay term, and $\beta \in (0,\infty)$ be the singular value threshold above which we start applying the decay term. Additionally, suppose that the weight updates are constrained to have norm $||\Delta W|| \leq \eta$ such as with the Muon optimizer. Then spectrally clipped weight decay has an equilibrium point,*

$$\sigma_{eq} = \beta + \frac{1-\lambda}{\lambda}\eta, \tag{12}$$

*toward which it "pulls" the spectral norm of the weights.*

*Proof.* An update step yields

$$W_{t+1} = \text{spectral\_clipped\_weight\_decay}_{\lambda,\beta}(W_t + \Delta W_t).$$

The subadditivity of norms tells us $||W_t + \Delta W_t|| \leq ||W_t|| + ||\Delta W_t|| \leq ||W_t|| + \eta$. Thus, we can bound the spectral norm of the weights after every update step as

$$\sigma'_{\max} \leq \mathsf{clipped\_weight\_decay}_{\lambda,\beta}(\sigma_{\max} + \eta)$$

$$\sigma'_{\max} \leq \begin{cases} \sigma_{\max} + \eta & \texttt{if} \ \ \sigma_{\max} + \eta \leq \beta \\ (1 - \lambda)(\sigma_{\max} + \eta) + \lambda\beta & \texttt{if} \ \ \sigma_{\max} + \eta > \beta \end{cases}$$

Equality is achieved at $\sigma_{\mathrm{eq}}$, where

$$\sigma_{\mathrm{eq}} = \begin{cases} \sigma_{\mathrm{eq}} + \eta & \texttt{if} \ \ \sigma_{\mathrm{eq}} + \eta \leq \beta \\ (1 - \lambda)(\sigma_{\mathrm{eq}} + \eta) + \lambda\beta & \texttt{if} \ \ \sigma_{\mathrm{eq}} + \eta > \beta \end{cases}$$

$$\sigma_{\mathrm{eq}} = (1 - \lambda)\sigma_{\mathrm{eq}} + (1 - \lambda)\eta + \lambda\beta$$

$$\sigma_{\mathrm{eq}} = \beta + \frac{1 - \lambda}{\lambda}\eta$$

Note that singular values larger than $\sigma_{\mathrm{eq}}$ decrease after every update step,

$$\mathrm{update}(\sigma_{\mathrm{eq}} + \epsilon) = (1 - \lambda)(\sigma_{\mathrm{eq}} + \eta + \epsilon) + \lambda\beta$$

$$= \underbrace{(1 - \lambda)(\sigma_{\mathrm{eq}} + \eta) + \lambda\beta}_{\sigma_{\mathrm{eq}}} + (1 - \lambda)\epsilon$$

$$\mathrm{update}(\sigma_{\mathrm{eq}} + \epsilon) < \sigma_{\mathrm{eq}} + \epsilon,$$

since $1 - \lambda < 1$, while singular values smaller than $\sigma_{\mathrm{eq}}$ increase,

$$\mathrm{update}(\sigma_{\mathrm{eq}} - \epsilon) = (1 - \lambda)(\sigma_{\mathrm{eq}} + \eta - \epsilon) + \lambda\beta$$

$$= \sigma_{\mathrm{eq}} - (1 - \lambda)\epsilon$$

$$\mathrm{update}(\sigma_{\mathrm{eq}} - \epsilon) > \sigma_{\mathrm{eq}} - \epsilon.$$

Hence $\sigma_{\mathrm{eq}}$ is an equilibrium point. $\qquad\square$

A potentially useful property of spectrally clipped weight decay is that its equilibrium point approaches $\beta$ as learning rate is decayed to zero during training, independent of the chosen initial learning rate and decay term:

$$\sigma^*_{\mathrm{eq}} = \lim_{\eta \to 0} \beta + \frac{1 - \lambda}{\lambda}\eta = \beta.$$

This property may enable tighter final norm bounds without requiring as aggressive of a decay.

# D    PROVING AN UPPER BOUND ON THE LIPSCHITZ CONSTANT OF A TRANSFORMER

We elaborate on the algorithm sketched in Section 4.2 and prove a Lipschitz bound on attention. Our Lipschitz bounds are with respect to the max RMS norm over token positions, denoted $\|\cdot\|_{\infty\text{RMS}}$.

Recall the two primary ways Lipschitz constants $L_f$ and $L_g$ of two functions $f$ and $g$ interact:

- **Adding:** $f + g$ has Lipschitz constant at most $L_f + L_g$.
- **Composing:** $f \circ g$ has Lipschitz constant at most $L_f \cdot L_g$.

**Step 1: Residual connections.** Suppose that, before reaching a certain residual connection, a transformer maps input data $x$ to $f(x)$ with Lipschitz constant $L$. Suppose the transformer has $2N$ residual connections. Let $\alpha = \frac{1}{2N}$. The residual connection acts on $f(x)$ as

$$[(1 - \alpha) \cdot \mathsf{identity} + \alpha \cdot \mathsf{block}]\,(f(x)). \tag{13}$$

After the residual connection, the Lipschitz constant composes and adds to become at most

$$(1 - \alpha) \cdot L + \alpha \cdot L \cdot L_{\mathsf{block}}. \tag{14}$$

Applying this formula sequentially upper bounds the Lipschitz constant of a transformer layer by layer. We now determine $L_{\mathsf{block}}$ for an MLP and attention block in terms of their weight norms.

**Step 2: MLP.** Our MLP composes $W_{\mathsf{out}} \circ (\mathsf{GeLU}/1.1289) \circ W_{\mathsf{in}}$. The Lipschitz constants of the two weight matrices are their norms $\|W_{\mathsf{out}}\|_{\mathrm{RMS}\to\mathrm{RMS}}$ and $\|W_{\mathsf{in}}\|_{\mathrm{RMS}\to\mathrm{RMS}}$, while $\mathsf{GeLU}/1.1289$ has Lipschitz constant 1 because we divide by the maximum derivative of GeLU. Overall, the Lipschitz bound for an MLP block is $L_{\mathsf{MLP}} \leq \|W_{\mathsf{out}}\|_{\mathrm{RMS}\to\mathrm{RMS}}\|W_{\mathsf{in}}\|_{\mathrm{RMS}\to\mathrm{RMS}}/1.1289$.

**Step 3: Attention.** Let $\ell$ denote the token dimension. Let the queries, keys, and values be denoted by $(q, k, v) \in \mathbb{R}^{\ell \times d_Q} \times \mathbb{R}^{\ell \times d_Q} \times \mathbb{R}^{\ell \times d_V}$. Our attention block composes

$$\tfrac{1}{3} W_{\mathsf{out}} \circ F, \tag{15}$$

where function attention is denoted by $F = \mathrm{softmax}\left(\frac{1}{d_Q} q k^\top + M\right) v$ for some mask $M$. As a consequence of the following theorem, if every attention input is unit norm, then functional attention is 1-Lipschitz. This property is what motivates scaling functional attention by $\frac{1}{d_Q}$ rather than $\frac{1}{\sqrt{d_Q}}$ inside the softmax. Composing functional attention with its input, the tuple $(q, k, v)$, increases its sensitivity to 3; we scale by $\frac{1}{3}$ to make attention as a whole have unit sensitivity. However, functional attention is no longer 1-Lipschitz if its inputs are not unit norm. Recall that the shorthand notation $\|x\|_{\infty\text{RMS}}$ is the max RMS norm of a $d$-dimensional activation over $l$ tokens, $x \in \mathbb{R}^{\ell \times d}$.

**Theorem D.1** (Lipschitz bound on functional attention). *Let $\diamond$ denote tensor contraction. Given any perturbations $\Delta q, \Delta k, \Delta v$ to the queries, keys, and values, functional attention satisfies*

$$\|\nabla F(q, k, v) \diamond (\Delta q, \Delta k, \Delta v)\| \leq \max(1, \|v\| \max(\|q\|, \|k\|))\|(\Delta q, \Delta k, \Delta v)\|, \tag{16}$$

*where the norm is $\|\cdot\|_{\infty\text{RMS}} : \mathbb{R}^{\ell \times d} \to \mathbb{R}$, the max-over-tokens* RMS *norm of the embedding vector, and $\|(\Delta q, \Delta k, \Delta v)\| := \|\Delta q\| + \|\Delta k\| + \|\Delta v\|$. That is, functional attention has Lipschitz bound $\max(1, \|v\| \max(\|q\|, \|k\|))$.*

*Proof.* The argument mirrors the proof of Proposition 7 from the modular norm paper (Large et al., 2024). We write the attention matrix as $A = \mathrm{softmax}\left(\frac{1}{d_Q} q k^\top + M\right)$. Its derivative is $\Delta A = \nabla_{(q,k)}\mathrm{softmax}\left(\frac{1}{d_Q} q k^\top + M\right) \diamond (\Delta q, \Delta k)$. The derivative of $F$ splits into two terms,

$$\nabla F(q, k, v) \diamond (\Delta q, \Delta k, \Delta v) = A(\Delta v) + (\Delta A)v. \tag{17}$$

We call the maximum $\ell_1$ norm of the rows of a matrix its $L^\infty$ operator norm, which comes into play by observing that $\|Ax\|_{\infty\text{RMS}} \leq \|A\|_{\infty-\mathsf{op}}\|x\|_{\infty\text{RMS}}$. For the first term, note that $\|A\|_{\infty-\mathsf{op}} = 1$ because softmax ensures the row-wise sum is always 1. For the second term, Large et al. (2024) in Equation E.58 show that

$$\|\Delta A\|_{\infty-\mathsf{op}} \leq \|\Delta q\|_{\infty\text{RMS}}\|k\|_{\infty\text{RMS}} + \|q\|_{\infty\text{RMS}}\|\Delta k\|_{\infty\text{RMS}}. \tag{18}$$

Thus, writing $\|\cdot\|$ as shorthand for $\|\cdot\|_{\infty\mathrm{RMS}}$,

$$
\begin{aligned}
\|\nabla F(q,k,v) \diamond (\Delta q, \Delta k, \Delta v)\| &= \|A(\Delta v)\| + \|(\Delta A)v\| \\
&\leq \|A\|_{\infty-\mathsf{op}}\|\Delta v\| + \|\Delta A\|_{\infty-\mathsf{op}}\|v\| \\
&\leq \|\Delta v\| + \|v\|\|k\|\|\Delta q\| + \|v\|\|q\|\|\Delta k\| \\
&\leq \|\Delta v\| + \|v\| \max(\|q\|, \|k\|)(\|\Delta q\| + \|\Delta k\|) \\
&\leq \max(1, \|v\| \max(\|q\|, \|k\|))(\|\Delta q\| + \|\Delta k\| + \|\Delta v\|)
\end{aligned}
$$

Hence, $\|\nabla F(q,k,v) \diamond (\Delta q, \Delta k, \Delta v)\| \leq \max(1, \|v\| \max(\|q\|, \|k\|))\|(\Delta q, \Delta k, \Delta v)\|$ as claimed. $\square$

More generally for attention layers with attention scale $s_{\mathsf{attn}}$ not necessarily equal to $\frac{1}{d_Q}$, we can absorb the extra factor to the query weight $W_Q$ *and* key weight $W_K$, such that

$$
\tilde{F} = \mathrm{softmax}\left(s_{\mathsf{attn}}qk^T + M\right)v = \mathrm{softmax}\left(\frac{1}{d_Q}\tilde{q}\tilde{k}^T + M\right)v, \tag{19}
$$

where $\tilde{q} = \sqrt{s_{\mathsf{attn}}d_Q}\,q$ and $\tilde{k} = \sqrt{s_{\mathsf{attn}}d_Q}\,k$. The Lipschitz bound then is,

$$
\max(1, \|v\| \max(\|\tilde{q}\|, \|\tilde{k}\|)) = \sqrt{s_{\mathsf{attn}}d_Q} \max(1, \|v\| \max(\|q\|, \|k\|)). \tag{20}
$$

**Step 4. Activation norm bounds.** To apply the theorem, we now bound the input norm to attention. To do so we will track the maximum RMS norm of activations everywhere in the network. We do not use layer norm and therefore cannot reset activation norms to 1. Let $x_0, \ldots, x_{2N}$ denote all the activations, from the initial embedding $x_0$ through to the $N$ alternating attention and MLP blocks acting via residual connections. Suppose the embedding layer maps tokens to have RMS norm at most 1, or $\|x_0\|_{\infty\mathrm{RMS}} \leq 1$. Attention and MLP increase the norm as follows:

- Attention computes $W_{\mathsf{out}} \circ (V, A)$ for some attention matrix $A$, where $(V, A)$ is shorthand for functional attention. By definition $V$ cannot increase the RMS norm of the embedding $x_i$ at any token by more than its RMS $\rightarrow$ RMS operator norm, meaning $\|Vx_i\|_{\infty\mathrm{RMS}} \leq \|V\|_{\mathrm{RMS}\rightarrow\mathrm{RMS}}\|x_i\|_{\infty\mathrm{RMS}}$. The same bound applies to $(V, A)x_i$ by subadditivity of norms, since entries of the attention matrix $A$ sum to 1 in the token dimension. Therefore attention can increase the activation norm by

$$
\|(W_{\mathsf{out}} \circ (V, A))x_i\|_{\infty\mathrm{RMS}} \leq \|W_{\mathsf{out}}\|_{\mathrm{RMS}\rightarrow\mathrm{RMS}}\|V\|_{\mathrm{RMS}\rightarrow\mathrm{RMS}}\|x_i\|_{\infty\mathrm{RMS}}. \tag{21}
$$

  In words, multiply the weight norms of $W_{\mathsf{out}}$ and $V$ to get the maximum increase.

- The MLP computes $W_{\mathsf{out}} \circ (\mathsf{GeLU}/1.1289) \circ W_{\mathsf{in}}$. Therefore the MLP can increase activation norm by $\|W_{\mathsf{out}}\|_{\mathrm{RMS}\rightarrow\mathrm{RMS}}\|W_{\mathsf{in}}\|_{\mathrm{RMS}\rightarrow\mathrm{RMS}}/1.1289$, since $|\mathsf{GeLU}(x)| \leq |x|$ for all $x \in \mathbb{R}$.

- The residual connection acts like

$$
\|(1-\alpha)\cdot x_i + \alpha\cdot\mathsf{block}(x_i)\|_{\infty\mathrm{RMS}} \leq (1-\alpha)\|x_i\|_{\infty\mathrm{RMS}} + \alpha\|\mathsf{block}(x_i)\|_{\infty\mathrm{RMS}}. \tag{22}
$$

**Algorithm to compute Lipschitz bound.** Therefore, given the weight norms of all matrices in a transformer, we use the preceding results to compute its Lipschitz bound in two steps. First, we upper bound the activation norm everywhere in the network using Step 4. Second, we upper bound the Lipschitz constant using Steps 1-3. The Lipschitz bound after the final layer is what we refer to as the transformer's Lipschitz upper bound.

# E  IMPLEMENTING LIPSFORMER AND BOUNDING ITS LIPSCHITZ CONSTANT

To turn our enforced norm training into LipsFormer (Qi et al., 2023), we make the following changes:

1. Remove spectral soft cap and embed projections.

2. Use CenterNorm: mean subtraction with learnable entrywise scale and bias.

3. Use scaled-head cosine attention with $\epsilon = 10^{-6}$, $\tau = 12$, $\nu = 1$. Notably, the official implementation of LipsFormer uses $\epsilon = 0$. According to their Theorem 1, this choice may make a finite Lipschitz bound impossible. We set $\epsilon > 0$ to fix the issue.

4. Heuristically scale down attention output by $1/n_{\text{heads}}$ to match their implementation.

5. Insert residual connections with learnable strength $\alpha$, initialized to $1/n_{\text{residual\_connections}}$.

6. Xavier normal initialize linear layers, then apply spectral normalization $W \mapsto W/\|W\|_*$.

7. Include drop path: every residual connection is skipped with $p = 0.5$ and, if taken, is scaled up by $1/(1 - p)$, matching their official implementation which uses `nn.Dropout`.

8. Use weight decay 0.1, matching their implementation (not applied to scalar parameters).

9. Use the Muon optimizer to give LipsFormer the fairest comparison, copying hyperparameters from our run. We tested training with AdamW for all parameters, an exact replication, but found performance degraded sigificantly: after 1770 steps, validation loss was 4.86 (compared to 3.61) and validation accuracy was 0.227 (compared to 0.301).

10. For non-weight-matrix parameters, use Adam hyperparameters $\eta = 0.001$, $\beta_1 = 0.9$, $\beta_2 = 0.999$, $\epsilon = 10^{-8}$ to match their implementation.

11. Use cosine learning rate schedule with decay to 0 to match their implementation.

**Bounding the Lipschitz constant of LipsFormer.** In Table 1, we report that our trained implementation of LipsFormer has a Lipschitz upper bound of $10^{130}$. To calculate this value, we use the final weight norms of the MLP and attention blocks to bound the Lipschitz constant of each residual block, relying on LipsFormer's Theorem 1:

$$\mathsf{Lip}(\mathrm{SCSA})_2 \le 2N(N-1)\nu\tau\epsilon^{-\frac{1}{2}}\|W^K\|_2 + 2(N-1)\nu\tau\epsilon^{-\frac{1}{2}}\|W^Q\|_2 + 2N\nu\epsilon^{-\frac{1}{2}}\|W^V\|_2.$$

Using $N = 128$ (head dimension), $\tau = 12$, $\nu = 1$, and empirical weight norms, we calculate the Lipschitz bound for every layer. We use the maximum entry of the learned residual strength $\alpha$, which is an entrywise multiplication, to convert the layerwise bounds into a final bound

$$\mathsf{Lip}(F) \le \prod_{s=1}^{S} \prod_{m=1}^{S} (1 + \alpha_{s,m}\mathsf{Lip}(f_{s,m})),$$

which we take from their Equation 19. Alpha has typical maximum entries around 0.5 for attention connections and 0.15 for MLP connections. With $\epsilon = 10^{-6}$, we compute a final Lipschitz bound of $1.97 \times 10^{129}$.

## F EXPERIMENTAL DETAILS

This section gives experimental details for all results in the paper. The three categories of experiments we run are MLP training, Shakespeare transformer training, and NanoGPT speedrun training.

**Datasets.**

- For MLP training we use the CIFAR-10 dataset Krizhevsky (2009) with the standard train and test splits and no data augmentation. We do not shuffle the order of batches.

- For Shakespeare transformer training we use Karpathy's 1M character-level dataset with standard training and validation splits (Karpathy, 2022). We shuffle the order of batches.

- For NanoGPT speedrun transformer training we use the FineWeb10B dataset (Penedo et al., 2024) loaded in the standard order. We use the same validation split as the modded NanoGPT speedrun benchmark (Jordan et al., 2024a).

**Compute requirement.** All our experiments can run on a V100, A100, or H100 GPU in less than 5 minutes, except the NanoGPT speedrun transformer which requires 8xH100 and runs in 5-10 minutes.

**Modula library.** For MLP and Shakespeare experiments, we use JAX (Bradbury et al., 2018) on top of the Modula library (Large et al., 2024; Bernstein, 2025). We implement our own model components. Our AdamW implementation does not include bias correction, although the discrepancy decays rapidly after aronud 20 steps because we use $\beta_1 = 0.9, \beta_2 = 0.95$ in all experiments except one, not reported, in which we determine that this is a good setting for the momentum EMAs.

**MLP experiments.** All MLPs we train are width 256 and depth 3 (i.e., one hidden layer) with ReLU activations and no bias on data from CIFAR-10. We use batch size 512 and a linear learning rate schedule that decays to 0 in all experiments. Modula's mass calculation causes the effective learning rate to be scaled by $1/3$. We train for 50 epochs except in one case, when we train for 20 epochs for the models in Figure 3. We zero-initialize the final layer. We train all models in float32 precision and run the weight constrain methods in float32 precision. We experimented with lower precision and found comparable metrics across the board for bfloat16 training. We set seed 0 and store all hyperparameters and log information to enhance reproducibility.

**Shakespeare experiments.** All transformers we train for Shakespeare are width 256 with 3 blocks (attention + MLP), no bias, and four attention heads. The out projection in each attention and MLP block is initialized to zero. We use sequence length 256 and batch size 64 to match the baseline from (Karpathy, 2022), except we train for 2000 steps while Karpathy trains for 5000 steps. We set Modula's blocks mass parameter to 32 to cause 95% of the feature learning to occur in the transformer blocks. We determined this ratio by sweeping the blocks mass, which controls the ratio of learning rate between the two embedding layers and the transformer blocks. Training with Muon means applying Muon to all linear layer weight matrices (including the final logit head) but normalizing the columns of embedding gradient, as suggested by the $\ell_1 \to \text{RMS}$ duality map (Bernstein & Newhouse, 2025). We were concerned that rare tokens may cause the momentum buffer to dualize columns to full strength updates for hundreds of steps until the column decays to exactly zero, so we tested whether capping the maximum inflation factor for the embedding column normalization could help. We tested maximum factors in the set $\{1, 4, 16, \dots, 65536\}$ across 8 seeds and found no significant difference. We choose to maximally multiply each column by 16 during the dualization step. Finally, we found that to train to the validation losses reported we had to use a trick: we decayed the learning rate by a factor of 1/2 per residual layer, causing later layers to train more than earlier layers. This change is implemented by setting the sensitivity of the Mul module in Modula to 1. We do not know why this trick is necessary.

Figure 2 sweeps over the following hyperparameters, following a coordinate descent hyperparameter search method:

- MLPs on CIFAR-10: we test the following combinations of optimizer and weight constraint method: (AdamW, weight decay), (AdamW, spectral weight decay), (AdamW, spectral normalization), (AdamW, Stiefel manifold projection), (AdamW, spectral hammer), (Muon, weight decay), (Muon, spectral weight decay), (Muon, spectral normalization), (Muon, stiefel manifold projection), (Muon, spectral hammer), (Muon, spectral soft cap), (Muon, spectral hard cap). For AdamW, we vary the weight decay and spectral weight decay

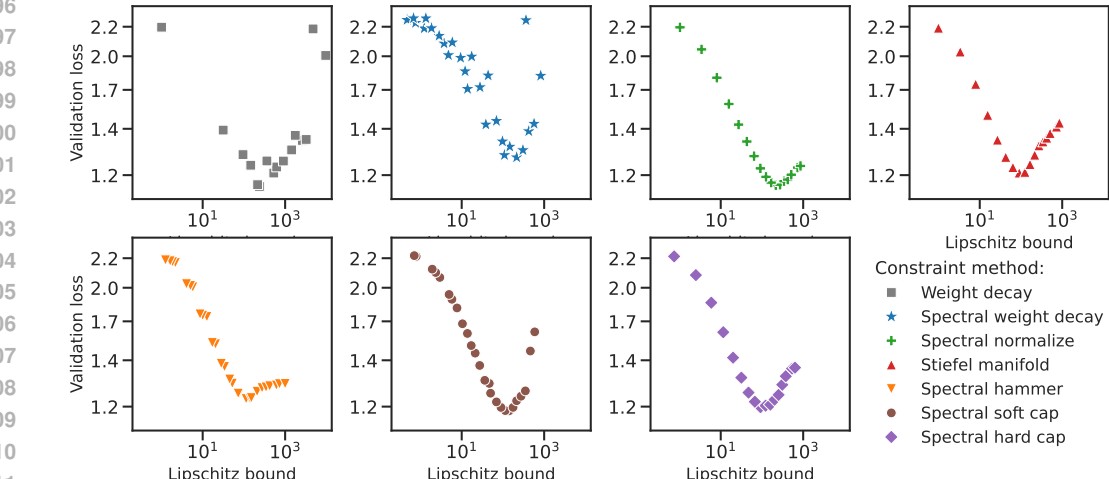

Figure 6: **Lipschitz vs. loss tradeoff broken down by method.** Each point shows the lowest validation loss achieved at a given bound for CIFAR-10 MLPs, split among methods, rather than aggregated as in Figure 4, left. We see clearly here that to remain on the frontier of Lipschitz vs. loss tradeoff (left hand of the parabola), it is best to use spectral capping, spectral hammer, or spectral normalization.

        parameters with 10 points in log-space from $10^{-2}$ to $10^0$. For Muon we vary the weight decay parameter with 10 points in log-space from $10^{-3}$ to $10^0$ and the spectral weight decay parameter with 10 points in log-space from $10^{-2}$ to $10^0$. For AdamW with spectral normalization, Stiefel manifold projection, and spectral hammer, we vary the maximum weight norm in the set $\sigma_{\max} \in \{2, 3, 4, 5, 6, 7, 8\}$. For Muon with spectral normalization, Stiefel manifold projection, spectral hammer, spectral soft cap, and spectral hard cap, we vary the maximum weight norm in the set $\sigma_{\max} \in \{1, 1.5, ..., 9.5, 10\}$. For AdamW with all methods we sweep 16 learning rates in log-space between $10^{-5}$ and $10^{-0.5}$. For Muon we sweep 16 learning rates in log-space between $10^{-2}$ and $10^1$. Overall, this results in 1,610 total combinations, 682 with AdamW and 2,703 with Muon.

- Transformers on Shakespeare: we test the following combinations of optimizer and weight constraint method: (AdamW, weight decay), (AdamW, spectral normalize), (AdamW, spectral hammer), (Muon, weight decay), (Muon, spectral normalize), (Muon, spectral soft cap). For spectral normalize, spectral hammer, and spectral soft cap, we vary the maximum weight norm in the set $\sigma_{\max} \in \{1.0, 1.2, \ldots, 2.8, 3.0\}$. For the baseline, we vary weight decay in the set $\lambda \in \{2/3, 0.5, 0.4, 0.3, 0.2, 0.1, 0.05, 0.03, 0.01, 0\}$. For AdamW we sweep 16 learning rates between $10^{-4.5}$ and $10^{-1.5}$. For Muon, we sweep 12 learning rates between $10^{-1.5}$ and $10^{1.5}$. We ran tests before to find ranges that cover the optimal learning rate.

Figure 3 reports adversarial examples and dataset-wide statistics from two models trained for 20 epochs. The AdamW model is trained with learning rate $8.1 \times 10^{-3}$ and weight decay $\lambda = 0.1$. The Muon model is trained with learning rate $2.3 \times 10^{-1}$ and weight decay $\lambda = 0$, using the spectral soft cap method with a weight constraint of $\sigma_{\max} = 3$.

The left panel of Figure 4 visualizes the same data from the experiment for Figure 2, but focuses only on MLPs trained with Muon on CIFAR-10. We break down this panel by method in Figure 6. The middle and right panels use the Muon optimizer, with the following tuples of (weight constraint method, maximum singular value, weight decay, spectral weight decay, learning rate): (weight decay, N/A, 0.1, 0, 1.585), (spectral weight decay, N/A, 0, 0.05, 0.157), (spectral normalization, 6, 0, 0, 1.0), (Stiefel manifold projection, 5, 0, 0, 1.0), (spectral hammer, 4, 0, 0, 0.398), (spectral soft cap, 6, 0, 0, 0.398), (spectral hard cap, 5, 0, 0, 0.631).

**NanoGPT experiments.** Following the Modded-NanoGPT speedrun standard (Jordan et al., 2024a), our training runs print log files with the full source code required to reproduce the results. We briefly

summarize the changes we made to convert the NanoGPT speedrun record (as of May 2025) into our method:

- Every step, RMS normalize the embedding columns.

- Initialize all linear layer weight matrices to be orthogonal.

- Reparameterize residual connections according to Equation (1): $\frac{L-1}{L}x + \frac{1}{L}\text{block}(x)$ residual connections, where $L = 24$ is the number of residual connections.

- Reparameterize attention according to Equation (2): $\frac{1}{3}$ overall scale on the attention output and $1/d_{\text{head}}$ scale inside the softmax.

- Every step, apply spectral soft cap (or spectral normalize) to every linear layer weight matrix based on a prespecified maximum desired weight norm $\sigma_{\text{max}}$.

- Use different orthogonalization coefficients that at most inflate a singular value to 1.14502. Therefore, the maximum update norm we pass to the strength parameter solver for learning rate coupling in spectral soft cap is $\eta \cdot 1.14502 \cdot 1.05$ with an extra factor of 1.05 to be safe around numerical precision errors. The iteration is derived by modifying the method in (Cesista et al., 2025).

- Remove U-net structure.

- Use GeLU$/1.1289$ instead of ReLU$^2$.

- Switch the dimension scaling in Muon to be $\sqrt{\text{fan\_out/fan\_in}}$ instead of $\max(1, \sqrt{\text{fan\_out/fan\_in}})$.

- Remove RMS normalization: the model is now Lipschitz continuous.

- Add back the 7th attention layer (which was removed in the speedrun).

- Weight projections are run in bfloat16 (which we found to slightly improve performance). Spectral normalization uses 2 iterations, meaning that weight norms can exceed the specified maximum $\sigma_{\text{max}}$ due to approximation error; in practice weights with norms enforced by spectral normalization exceed the specified maximum by around 10%.

