# OpenReview forum: "Training Transformers with Enforced Lipschitz Bounds"
_ICLR.cc/2026/Conference — Submitted to ICLR 2026_

### Official Review · Reviewer_QoE5 · 2025-10-27

**Soundness:** 3
**Presentation:** 1
**Contribution:** 3
**Rating:** 2
**Confidence:** 4

**Summary:**

This paper presents methods for training transformers with Lipschitz bounds enforced throughout training. The authors develop spectral soft cap and spectral hammer techniques, demonstrate that Muon optimizer works better than AdamW for Lipschitz-constrained training, and scale to 140M parameter transformers. However, competitive performance requires astronomically loose Lipschitz bounds (10^122), significantly undermining the practical utility.

**Strengths:**

The motivation of the paper is well-rooted and is the subject of reliability for AI systems.

The paper presents an interesting avenue of research regarding Lipschitz transformers.

The authors do explore several empirical avenues for their proposed methodology and build on recent work.

**Weaknesses:**

My first and foremost issue is that the paper does not establish the baselines and the necessary frameworks appropriately. Especially, as this work ventures into semi-rigorous mathematics, it is critical to establish the frameworks that this work builds on properly.
Critically, the optimizers that this paper talks about should be properly established, and their steps (or at least the relevant steps) must be clearly stated so that one can easily verify the claims of the authors and follow through the logic of the work. Despite my relatively deep knowledge of mathematical optimization, I do not have the steps of AdamW, or the very novel work Muon, in memory, and this is not acceptable for a paper that seeks to build upon these works and expand them, and study a phenomenon under these two different optimizers.
This is a concern that covers sections 3 and 4.

In my view, unless the paper fixes this issue so that it is easier to read and follow, it would not rise to the level of ICLR.

It is indeed new and somewhat odd that I see the use of the RMS norm in the literature of Lipschitz continuity. Nonetheless, this is not a big issue. However, it does cast some uncertainty on certain experimental results.
Firstly, it makes Figure 3 slightly harder to compare with existing baselines.
Secondly, it casts doubt on the Lipschitz bounds calculated for the LipsFormer in Tab1. Have the authors taken caution to make sure that the Lipschitz calculations are correct with respect to this new norm that the authors use?

I do not understand the focus of the authors on the activations. What is the significance of having small activations??
This question is regarding both the experiments and the main body.

Another point regarding activations is regarding eq (1). A residual connection as in eq 1 is always 1-Lipschitz, given block(x) is 1-lipschitz. Why do you need a bound on the activation norms???
I would also like to point the authors to works such as [5] that establish conditions on simple feed-forward residual connections that can be 1-Lipschitz with a non-modified residual connection. This can help the authors to make modifications to their text so that it is technically correct.


Finally, the experimental results of the paper seem subpar. I am aware that Lipschitz regularization reduces the natural accuracy of a classifier. However, the accuracy reported in this paper for CIFAR-10 is not acceptable. For example, adversarially trained CNNS [1] or resnets [2] present much higher accuracies on both $ l_2 $ and $ l_\infty $ perturbations, and certified trained methods such as those of [3] and [4] present better results. Thus, unless there is a significant difference between the setups, this is not acceptable.

The same argument follows for the results of Table 1, though I am less familiar with the baselines in this space. Unfortunately, a Lipschitz constant that surpasses the 1000s is potentially useless, and thus, the only relevant entry, I think, is the 10-Lipschitz model, which seems to have a significantly lower accuracy. Care to further explain why such a model could be useful?








[1] Zhang H, Yu Y, Jiao J, Xing E, El Ghaoui L, Jordan M. Theoretically principled trade-off between robustness and accuracy. InInternational conference on machine learning 2019 May 24 (pp. 7472-7482). PMLR.

[2] Xu Y, Sun Y, Goldblum M, Goldstein T, Huang F. Exploring and exploiting decision boundary dynamics for adversarial robustness. arXiv preprint arXiv:2302.03015. 2023 Feb 6.

[3] Fazlyab M, Entesari T, Roy A, Chellappa R. Certified robustness via dynamic margin maximization and improved lipschitz regularization. Advances in Neural Information Processing Systems. 2023 Dec 15;36:34451-64.

[4] Wang R, Manchester I. Direct parameterization of lipschitz-bounded deep networks. InInternational Conference on Machine Learning 2023 Jul 3 (pp. 36093-36110). PMLR.

[5] Araujo A, Havens A, Delattre B, Allauzen A, Hu B. A unified algebraic perspective on lipschitz neural networks. arXiv preprint arXiv:2303.03169. 2023 Mar 6.

**Questions:**

See weaknesses.

---

> ### Author Response · Authors · 2025-11-23
>
> **Dear reviewer QoE5**, thank you for your thoughtful and detailed evaluation. We address your major concerns below, and hope these clarifications help the clarity of our paper, raising it to a level worth acceptance. Please let us know if you have any additional questions in the discussion period. We appreciate your feedback, as it genuinely guided substantial improvements in the delivery of our results.
>
> **On establishing frameworks and clarity of optimizers.**
> We appreciate the reviewer’s emphasis on clarity. In the revision, we have added a dedicated section summarizing the steps of AdamW and Muon, including relevant update equations and the aspects that matter for our theoretical arguments. Specifically, we detail AdamW’s unconstrained update norm and Muon’s fixed spectral-norm update. We detail the order of operations clearly, so we more easily demonstrate our method first takes a standard optimizer step, then applies a constraint method after the weight update, as demonstrated in Fig. 1. These details are now provided directly in the main text so that readers do not need to recall them from memory or check outside papers to follow the logic in Section 3. We believe this substantially improves the readability and rigor of our mathematical framework.
>
> **On RMS norms and LipsFormer Lipschitz calculations.**
> We use $RMS \to RMS$ norms because they align with the modular-norm perspective used in work by Large et al., 2024 [1], which we use as a major inspiration for our work. $RMS \to RMS$ norms also make Lipschitz statements depth-scaled and dimension-independent. To address the reviewer’s concern about comparisons to $\ell_2$ baselines, we already explicitly note the conversion between $RMS \to RMS$ and $\ell_2$ Lipschitz bounds in Section 3.1, so readers can compare directly with prior work. As a reminder, you simply have to multiply by $\sqrt{d_{\text{output}} \/ d_{\text{input}}}$ for a given network. For our MLP experiments in Fig 2, this is a scale of $\sim 0\.057$, and for the transformer experiments in Fig. 2, the scale is $\sim 0\.50$. This means these bounds are all larger as $RMS \to RMS$ bounds than as $\ell_2$ bounds. This means in Fig. 3, the top MLP (Muon + soft cap) has $\ell_2$ Lipschitz bound $\sim 0\.87$ and the bottom MLP (Adam + weight decay) has $\ell_2$ Lipschitz bound $\sim 434\.3$. We have updated this figure to include both $RMS \to RMS$ and $\ell_2$ Lipschitz bounds. For table 1, we are reporting our own calculation of the LipsFormer baseline’s bound using consistent $RMS \to RMS$ conventions to ensure correctness. We added text clarifying this in the revision.
>
> **On the focus on activation norms.**
> Our emphasis on activation norms is not because small activation norms are a goal by themselves, but because they serve two important purposes for transformer training and one empirical signal for our results:
>
> 1. **Training stability in normalization-free transformers:** Without layer norm or QK norm, activation growth is an underlying cause of training divergence. Bounding activations is therefore essential for removing these stabilization tricks from transformers.
>
> 2. **Compatibility with low-precision training:** Constrained activation ranges allow us to train networks in fp8 precision without blockwise or tensorwise scaling. This is one practical benefit we identify (Section 5). We think major industry labs care a lot about quantization, and a recurring problem is if a single activation element (the max) is an outlier, then the uniform distribution over the tensor squishes all smaller values to zero. This degrades intelligence. Quantization is more accurate if the max(abs(activation)) is small, which is what we report. We are adding a better explanation of this to the paper!
>
> 3. **Evidence of empirically low Lipschitz constants:** One limitation of our work, which we address in Section 5, is that our bounds are compositional and while they are tighter than former theoretical bounds for transformer Lipschitz constants, they are likely loose. Therefore, for a representation to truly grow at the scale implied by our bound, it would have to perfectly align with the direction of the first principal vector at every layer of the network. This is extremely unlikely, so our bound in practice on real data is likely much lower than our theoretical bound. Our maximum activation norms give a better sense of the empirical reduction in norm that our methods allow us to achieve. To be clear, we think understanding the alignment and compositionality empirics better is an important direction to make progress on in the future.
>
> We have added clearer discussion on why activation magnitudes matter in transformer architectures and why they give us relevant signal for deploying these models.
>
> (1/2)

---

> > ### Author Response · Authors · 2025-11-23
> >
> > **On residual connections and activation bounds.**
> > Thank you for flagging this. The question you had about the activation norms is due to our poor wording, which we fixed. The crux is whether block(x) is 1-Lipschitz. Attention has a Lipschitz bound that depends on its input activation norms; that’s why we said the 1-Lipschitz bound on block(x) fails if the activations exceed 1. Obviously if one knew block(x) were 1-Lipschitz activations wouldn’t matter, but the Lipschitzness depends on activations in attention. Please see Theorem D.1 for details. Thank you for the suggested reference [5] which we have incorporated. We have made the text more precise here.
> >
> > **On CIFAR-10 results and comparison to existing frameworks.**
> > We agree that our reported CIFAR-10 accuracies are lower than convolutional architectures with adversarial training or certified robustness techniques. However, the settings are not comparable. In the experiments you refer to (Section 3.2), we are using three-layer MLPs with hidden dimension 256, not convolutional networks. We have stated this both in Section 3 and Appendix F. An apples-to-apples comparison with other shallow MLPs on CIFAR-10 in the literature confirms 60% accuracy is common and even high. Again we opted for a simple setup as a warmup and intuition building exercise, rather than attempting to max out the CIFAR-10 accuracy score (which is largely a solved problems—you can train to 95% accuracy in 3 seconds on an A100—see CIFAR-10 Airbench by Keller Jordan, it’s cool!). Our objective in Section 3 is to study weight-norm constraint methods for linear layers, the components used in transformers. The CIFAR-10 experiments serve as controlled, low-cost settings for comparing constraint settings and optimizer interactions, not competitive image classification performance overall. To make this clearer, we now explicitly state in Section 3 that the MLP experiments are diagnostic experiments intended to guide the development of our transformer methods.
> >
> > **On the usefulness of the <=10-Lipschitz transformer in Table 1.**
> > None of the models we trained are useful in a production setting (loss $\sim 3\.3$ on internet text is GPT-2 level). We view the utility as (1) defining a tradeoff frontier, (2) adding a new point that was previously impossible to train on to that frontier [the low Lipschitz point], and (3) encouraging the community to push the tradeoff frontier. This is an early paper in the green field of training transformers with enforced Lipschitz constants, but we can envision a future where production language models are trained stably with Lipschitz constant < 10, at which point this data point from our paper will look like GPT-1: not useful on its own, but indicating a method that can scale. At least, we hope this is the case. In research, one can never be sure. All we concretely do with our <=10-Lipschitz transformer is unlock a new point on the tradeoff frontier.
> >
> > Overall, we hope these address some of your concerns while making the technical contribution of our paper more clear. Please let us know whether you have any additional questions or concerns.
> >
> > (2/2)

---

### Official Review · Reviewer_3yVB · 2025-10-30

**Soundness:** 1
**Presentation:** 1
**Contribution:** 2
**Rating:** 2
**Confidence:** 3

**Summary:**

The paper studies training transformers with small Lipschitz constants with two optimizers: AdamW and Muon. To enforce a strong Lipschitz constraint without losing expressivity too much, the authors propose two new techniques: spectral soft capping (for Muon) and spectral hammer (for AdamW). Spectral capping works by reducing all singular values of the weight matrices with stronger effects for the larger singular values. Spectral hammer works by targeting the highest singular value in each of the weight matrices (estimated through power iteration) and setting it below a certain threshold. Since spectral hammer only targets one singular value at a time, it is more suitable for weight updates that exhibit low rank (and hence most singular values are close to zero anyways) which is the regime that Adam mostly operate in. This is not the case for Muon which encourages the update to stay high-rank, where soft spectral capping (which brings down all singular values at the same time) becomes more suitable. Empirically, both versions of  spectral capping (soft and hard) work effectively at maintaining a low Lipschitz bound while achieving high validation accuracy/low validation loss on FineWeb10B and Tiny Shakespear.

**Strengths:**

- The paper contains thorough empirical evaluations across over a wide range of Lipschitz bounds, allowing a fair assessment of the robustness/expressiveness trade-off of the proposed methods and other Lipschitz-regularization methods like spectral normalization

- The proposed methods are well-motivated and allow for low Lipschitz bounds while maintaining reasonable validation accuracy/loss.

**Weaknesses:**

- The paper does not contain enough information (especially in the main text) for me to fully assess the technical validity of the proposed methods. To name a few issues,

  - The details on how spectral hard capping is implemented are missing in the main body. The paper only says ``spectral hard cap uses the matrix sign function to approximate $\sigma \rightarrow \min(\sigma_{\mathrm{max}}, \sigma)$ on the singular values''. Matrix sign function is not defined anywhere/missing a reference. I am guessing it means $\mathrm{msign}(M) = U\mathrm{sign}(\Sigma) V^\top $ where $M=U\Sigma V^\top$, but it is never explicitly stated or referenced anywhere. It would be good if the authors could clarify this in more details.

  - To figure out what spectral hard capping is, I read through most of Appendix C, which was referenced from the main body. It was still unclear after this point what spectral hard capping does exactly. It seems that at L938 (in the appendix), the authors referenced Cesista (2025) for the implementation details of spectral clipping (a generalization of spectral hard capping), which leads to a blogpost. After examining this reference material, the blogpost seems to propose the same method as done in the paper (\emph{e.g.}, spectral hard capping). This raises not only clarity concern but also novelty concern.

- The proposed methods do not seem to push the Lipschitz vs. loss frontier. From both Figure 2 (transformer), 4, and Table 1, the standard spectral normalization method and the prior Stiefel manifold projection method seem to cover the same frontier achieved by the proposed methods. While it is nice to see a comprehensive study of different Lipschitz-constrained methods, it remains unclear to me what the significance of the proposed methods is.

**Questions:**

(1) In Appendix C.1, Line 989 to Line 992, why is $\mathrm{msign}(\beta\mathrm{msign}(W) - W) = U\mathrm{sign}(\beta I - \Sigma)V^\top $? When I tried to verify it, I could only get to the following step:

$$
\begin{align}
    &\mathrm{msign}(\beta\mathrm{msign}(W) - W)\\\\
    &= \mathrm{msign}(\beta U\mathrm{sign}(\Sigma)V^\top - U\Sigma V^\top)\\\\
    &= U\mathrm{sign}(\beta \mathrm{sign}(\Sigma) - \Sigma)V^\top
\end{align}
$$

Perhaps I am missing something here, but this seems to be only true when $\Sigma$ has strictly positive singular values, which is not in the assumption as far as I understood. Same thing applies for the second term (i.e., $\beta\mathrm{msign}(W) - W = U\mathrm{sign}(\beta I - \Sigma) V^\top$

(2) In Table 1, spectral normalize is studied extensively. How well does your proposed methods (e.g., spectral capping) do for $\sigma_{\mathrm{max}}=1$ compared to spectral normalize?

---

> ### Author Response · Authors · 2025-11-23
>
> **Dear Reviewer 3yVB**, we are grateful that you clearly spent a lot of time understanding the details of our paper and thinking thoroughly about how we can improve the clarity of our methods and mathematics. We have updated our paper with many of your suggestions, as we will detail below. First of all, we want to immediately address your novelty concern.
>
> **On spectral hard capping clarity and novelty.**
> Our manuscript was first released as a preprint. The blog post (Cesista 2025) presents an exposition of the same idea, including an implementation inspired by our preprint. Because the post offered a pedagogical explanation, we cited it in the appendix and removed some low-level implementation detail, which created an unnecessary dependence on an external source. In the revision, we have restored these details directly into Appendix C and clarified the timeline to avoid any confusion about originality. The blog post is now referenced only as an additional resource to help explain our method. We are sorry for not making this clear in our original draft. To be explicit, the authors of this paper invented spectral hardcapping while the blog post (Cesista 2025) came afterward.
>
> Furthermore, you are completely right that we forgot to define the msign function. We have added that it is the function $W \= U S V^T \to U V^T$ where $U S V^T$ is the singular value decomposition of the matrix $W$. We will add this note to the main body and detail it a bit more in the appendix.
>
> Next, we will answer your questions.
>
> **In appendix C.1 … Same thing applies for the second term.**
> You are correct that singular values must be strictly positive for this to be true. In Machine learning workloads, this is almost always the case, unless we are dealing with a down projection where zero singular values stay zero after msign. It may help to reason through it by imagining that $\beta \mathrm\{msign\}\(W\) \- W$ is $U \(\beta I \- S\) V^T$ when $\mathrm\{msign\}\(W\) \= U V^T$, i.e., all singular values are nonzero, so $S \to I$. This is an important condition to state explicitly, and we have added it to appendix C.
>
> **In Table 1 … compared to spectral normalize?**
> As we show in Table 1, spectral normalization with $\sigma_{\max} \= 1$ results in accuracy 0.212 and loss 5.047. With spectral soft capping and $\sigma_{\max} \= 1$, we get accuracy 0.214 and loss 5.400. Our bound of 10 is the same.
>
> **On whether the proposed methods advance the Lipschitz–loss frontier.**
> Thank you for this thoughtful comment. We agree that spectral normalization is a strong baseline, and indeed Figure 2 and Figure 4 show that it lies on and sometimes even defines the Lipschitz–loss frontier. This is an important result in its own right: our study establishes spectral normalization, when paired with Muon, as a highly effective constraint method for transformer training. We view our research as advocating for weight constraints in general and not that our particular method (spectral capping) is best, although it performs reasonably well. We actually think it is exciting that a well-known and simple method (spectral normalization) pairs well with Muon.
>
> We would like to clarify two points.
>
> **First**, Stiefel manifold projection does not achieve a similar frontier in our experiments. Across both MLPs and transformers, Stiefel projection consistently underperforms spectral normalization and spectral capping in the low-Lipschitz, low-loss regime. In Figure 4, for example, Stiefel projection appears only in higher loss regimes of the frontier and does not reach within 1% accuracy of the baseline except at substantially larger bounds. Thus, while Stiefel is an important prior technique, our experiments do not show it covering the same frontier as our methods and spectral normalization.
>
> **Second**, our contributions include identifying new optimizer–constraint interactions. We view the paper’s significance as partly empirical: it reveals that the choice of optimizer (AdamW vs. Muon) fundamentally changes the achievable Lipschitz–loss tradeoff for classic methods such as spectral normalization and weight decay (Fig. 2). This observation had not been documented in prior work and has practical implications for Lipschitz-constrained training more broadly.
>
> Given that you seem to value our work as demonstrating the trade-offs between Lipschitz bounds and performance on reasonable tasks, we wonder whether you would consider elevating your rating of our paper to an acceptance after engaging in discussion.

---

> > ### Comment · Reviewer_3yVB · 2025-11-26
> >
> > Thanks for the clarifications on the definition of the matrix sign function and the method. Now I am able to understand both spectral softcapping and hardcapping and appreciate the their values. Since the work is mostly empirical, it is crucial to put key implementation detail about them in the main body and provide justifications. I think the paper could improve on this aspect (e.g., Section 3.1 only mentions hardcapping very briefly and does not explicit write out softcapping).
> >
> > Overall, I do think this work has a lot of potentials. However, right now the paper sort of situates at an awkward spot:
> >
> > (1) The paper presents a decently thorough empirical study on the Lipschitz/performance trade-off for Adam and Muon with various Lipschitz-constraining methods. While it is interesting to see how different optimizers behave under different Lipschitz-constraining methods, this alone does not provide actionable guidance for practitioners. Some additional analysis could uncover deeper/more impactful insights. Something that came to my mind (might not be the most promising direction as I am not as familiar in the field as the authors) is that perhaps there are some generalization/robustness differences between Adam and Muon or among Lipschitz-constraining methods.
> >
> > (2) The paper proposes new techniques (e.g., spectral softcapping/hardcapping/hammer) that are sound and practical, but the performance of these new techniques is a bit lacking at the moment. If there are particular advantages that these methods offer relative to prior methods (e.g., speed, generalization, robustness, implementation simplicity), highlighting them could make a stronger case why it is a good idea for the community to adopt them.
> >
> > I believe the paper could be much more impactful if either of the two is addressed. I am raising my score to 4 for now.

---

> ### Author Response · Authors · 2025-11-29
>
> Reviewer 3yVB, thank you very much for the thoughtful follow-up and for saying you would raise your score. We appreciate your engagement with the work! Below we address your suggestions and outline the concrete changes we are making in the revision.
>
> **Bringing key implementation details of softcapping and hardcapping into the main text.** We agree that, since the contribution is largely empirical, the clarity and accessibility of the methods in the main body are critical. In the revision we will expand Section 3.1 to explicitly write out both spectral softcapping and hardcapping and summarize the update rule and intuition for each method (with the detailed derivations remaining in Appendices).
>
> **Additional actionable insights for practitioners.** We agree that highlighting actionable takeaways makes the study more valuable. Based on your suggestion, we are adding a short section synthesizing results across Sections 3–4 to describe practical guidance, including: when Muon provides a clear benefit over AdamW (e.g., small-Lipschitz and strict-constraint regimes), the regimes where spectral normalization remains a strong default baseline, and the relationship between activation magnitudes, stability, and fp8 compatibility. We will also conduct a light additional analysis examining generalization and robustness differences between AdamW and Muon under matched Lipschitz budgets, similar to figure 3, but matching budgets rather than clean accuracy.
>
> Again, we appreciate your engagement and your constructive suggestions. We believe these revisions materially improve the clarity, usefulness, and impact of the paper, and we appreciate your engagement so far. We understand, unfortunately, that you won’t be able to interact any further due to the unfortunate circumstances surrounding the identity leak, and we enjoyed the brief discussion we were able to engage in before it.

---

### Official Review · Reviewer_rsDr · 2025-11-01

**Soundness:** 2
**Presentation:** 2
**Contribution:** 2
**Rating:** 4
**Confidence:** 3

**Summary:**

This paper develops new, computationally efficient tools—spectral capping and spectral hammer—to maintain norm-constrained weight matrices during training. These techniques enable the successful training of Lipschitz-bounded transformer models with up to 140M parameters. The authors also identify that the Muon optimizer plays a crucial role in achieving stability and improving the Lipschitz–performance tradeoff.

**Strengths:**

1.	This paper is well-organized and clearly written.
2.	The paper addresses a practical problem, enabling the training of Lipschitz-bounded transformers.

**Weaknesses:**

1.	There is a growing line of work on Lipschitz neural networks, which tightly bound the Lipschitz constant through architectural design or explicit Lipschitz regularization. These represent the mainstream direction in Lipschitz-bounded models, but the submission appears to omit key references, such as:

    [1] Hu, Kai, et al. "A Recipe for Improved Certifiable Robustness." ICLR, 2024.

    [2] Xu, Xiaojun, Linyi Li, and Bo Li. "Lot: Layer-wise orthogonal training on improving l2 certified robustness." NeurIPS, 2022.

2.	The contribution is somewhat limited. The proposed approach primarily leverages norm-normalization-based regularization of weight matrices, which may not meet the ICLR standard for conceptual novelty.
3.	The experimental section is relatively weak. The paper would benefit from more extensive experiments and in-depth discussion.
4.	The study focuses solely on language models. Including results on vision transformers would make the work more comprehensive and competitive.
5.	In Table 1, the results mainly show improved training stability without clear gains in performance. If the proposed methods merely stabilize training while sacrificing accuracy, the contribution appears insufficient. Since it is commonly understood that constraining the Lipschitz constant can enhance model robustness, the paper should discuss or empirically demonstrate this benefit (or others such as low-precision stability); otherwise, the practical significance of the proposed method remains unclear.

**Questions:**

1.	In Table 1, the condition number of the weight matrix using spectral capping or spectral hammer is not close to 1 but rather high. Does this lead to a looser Lipschitz bound?
2.	Standard methods often use layer normalization or QK normalization, whereas this paper adopts spectral-norm-based regularization. Is this understanding correct? If so, does the proposed norm-based method serve a similar role to layer norm or related normalization techniques?
3.	Has the paper considered incorporating the concept of orthogonality, where all singular values are equal to 1? Otherwise, the Lipschitz constant may grow with network depth. How does the proposed approach address this issue?

---

> ### Author Response · Authors · 2025-11-23
>
> **Dear Reviewer rsDr**, thank you for providing useful feedback for improving our paper. We appreciate your clear attention to detail as well as demonstrated knowledge of related methods, and hope we can address some of your main concerns.
>
> We will begin by answering your questions.
>
> **In Table 1… Does this lead to a looser Lipschitz bound?**
> Yes, using a w_max higher than one does increase our Lipschitz bound. For all of our methods, with w_max = 1, we achieve a network with Lipschitz bound 1, measured in $RMS \to RMS$ norm.
>
> **Standard methods often use… Is this understanding correct? If so, does the proposed norm-based method serve a similar role to layer norm or related normalization techniques?**
> Yes, your understanding is correct that we adopt spectral-norm based regularization, and that by doing this while aiming for certifiable Lipschitz bounds, we found this also allows us to remove layer normalization techniques. In this sense, our method does serve a similar role to layer norm, but also allows us to produce certified guarantees in adversarial robustness. In contrast, layer norm is not a Lipschitz operation because it can send $\pm \epsilon$ to $\pm 1$.
>
> **Has the paper considered incorporating the concept of orthogonality, where all singular values are equal to 1? Otherwise, the Lipschitz constant may grow with network depth. How does the proposed approach address this issue?**
> Yes, the method Stiefel Manifold Projection (Section 3.1) sets all singular values equal to 1. We will clarify that this is the same as orthogonalization in our update of the paper. Regarding the second part of your question, for any of our methods setting w_max = 1 prevents the Lipschitz constant from growing with network depth. This is because of our parameterization of residual connections within our transformer, described in Section 4.1.
>
> **Related work.**
> We thank the reviewer for pointing us to Hu et al. (2024) [1] and Xu et al. (2022) [2]. We agree these works are relevant and will incorporate them into our related-work discussion. Both papers investigate methods for enforcing Lipschitz constraints in convolutional architectures, developing tools that complement, but differ substantially from, our focus on constraining linear layers for transformer training. Hu et al. explore a broad set of spectral regularization techniques, while Xu et al. study smoothness control at the submodule level, conceptually related to our goal of shaping sensitivity but implemented in a different architectural setting. Because we target transformers rather than CNNs, their techniques are not directly applicable to our setting, but they provide valuable references for Lipschitz-bounded convolutional networks. We will revise the related-work section to acknowledge these contributions and clarify how our methods, spectral hammer and spectral capping, extend this line of work to transformer architectures.
>
> **On conceptual contribution**, our work does not apply existing norm-normalization techniques in order to develop new, theoretically-motivated mechanisms—spectral soft cap and spectral hammer—that efficiently enforce weight-norm constraints throughout training. We agree that in our transformer experiments, a majority of our results come from applying norm-normalization based methods. Section 3 introduces new analyses and Theorem B.1, which formally links Muon’s update structure to strict spectral-norm boundedness. We believe these contributions constitute meaningful conceptual novelty. We also show empirically (Fig. 2–4) that optimizer–constraint interactions fundamentally alter the Lipschitz–performance frontier, a phenomenon not previously documented. We will add more spectral soft-capping results to Table 1 when we update the paper.
>
> **Regarding the experimental section**, we agree that broader evaluation will strengthen the work. Our experimental design aimed to isolate the effect of weight-norm enforcement by performing extensive controlled sweeps (over 4k models across MLPs and transformers) to map the Lipschitz–performance tradeoff. We have expanded the discussion of these results in the revision to better highlight their significance, including the surprising finding that Lipschitz-enforced models can match or outperform baselines even under strict constraints. We saw some other comments about running more vision experiments. We think the large scale language experiments are sufficient, but if vision is important to you, we can do it too.
>
> (1/2)

---

> > ### Author Response · Authors · 2025-11-23
> >
> > **On evaluation beyond language models**, our initial focus on transformers for language was due to their sensitivity to training instabilities (attention logit growth, activation blow-up), making them a natural testbed for Lipschitz enforcement. However, the proposed weight-constraint mechanisms can apply directly to vision transformers. Under our current compute constraint, it is difficult to prepare additional ViT experiments. If you think they are important, we will try hard to add them to the revision. We appreciate the suggestion.
> >
> > **On training stability, accuracy, and practical significance.**
> > We appreciate the reviewer’s concern and agree that simply stabilizing training without demonstrable benefits would not constitute a sufficient contribution. Importantly, our work goes beyond stability alone in two ways.
> >
> > **First: our method enables training configurations that otherwise fail.** In Table 1, the unconstrained NanoGPT baselines diverge when layer norm, QK norm, and logit softcapping are removed, whereas our weight-constrained models train stably under these settings. This indicates that the proposed constraints enable training regimes that are otherwise impossible, providing a new path for normalization-free transformer training. This is an impact that is greater than merely stabilizing training.
> >
> > **Second: we empirically demonstrate meaningful benefits beyond stability.** Section 3.3 (Fig. 3) shows that, at matched clean accuracy, Lipschitz-bounded models require significantly larger $\ell_2$ perturbations to be fooled, and maintain higher accuracy across attack budgets. In Table 1, our 140M-parameter models maintain orders of magnitude smaller activation magnitudes (max entries 6.5–112) compared to baselines ($\sim 92k$ for NanoGPT). As discussed in section 5, these bounded activations allow all MLP and attention layers to run in fp8 without any per-tensor or per-block scaling, suggesting clear relevance for efficient and low-precision training.
> >
> > **Finally, we see our bounds as a knob that does not require sacrificing accuracy.** Notably this might require further tuning or adding more learnable parameters (such as soft cap as opposed to hard cap), which we see as fruitful future work. While imposing a strict Lipschitz bound can limit accuracy at extremely small budgets, our method lets practitioners choose the desired bound a priori and enforces it throughout training. For moderate bounds (e.g., w_max = 8 or 16), we match NanoGPT baseline performance while retaining the benefits above. These bounds are high, but still orders of magnitude lower than the unconstrained baselines.
> >
> > Overall, we hope these responses were helpful to your understanding of our paper. We genuinely appreciate you pointing us to new related work and offering perspectives that led to revisions throughout our paper. Given that you appreciated the contribution of our paper and clearly care about the endeavor of creating Lipschitz constrained models in machine learning, we hope you consider updating your score and championing our work as valuable to the body of knowledge that allows for scalable Lipschitz training across model architectures.
> >
> > (2/2)

---

### Official Review · Reviewer_WUi1 · 2025-11-03

**Soundness:** 3
**Presentation:** 3
**Contribution:** 3
**Rating:** 6
**Confidence:** 3

**Summary:**

This paper investigates the effects of enforcing Lipschitz bounds beyond initialization during the training of Transformer networks. Specifically, the paper removes activation normalization and constrains the weights throughout training, and finds that weight decay and spectral normalization are more effective with Muon than AdamW. Based on the conclusion, this paper further proposes spectral capping and spectral hammer to ensure the Lipschitz bounds during training. Comprehensive experiments are conducted to illustrate the impact of the technique on practical performance.

**Strengths:**

1. The impact of Lipschitz constraints during training of transformer-based networks is highly relevant for practical applications.
2. This paper is well-organized and easy to read.

**Weaknesses:**

1. More comprehensive experments on ImageNet-1K are needed to further demonstrate the generalization of the conclusion on Muon and AdamW.

**Questions:**

Please see the weakness.

---

> ### Author Response · Authors · 2025-11-23
>
> **Dear Reviewer WUi1**, thank you very much for taking the time to read our paper and provide feedback! We appreciate your excitement about the contribution and methodological updates. We are also glad to hear that you found our paper easy to read.
>
> We have one question regarding your weakness: *did you want us to do ImageNet-1k experiments with our current vision set up, which was with small MLPs, or were you imagining that we run experiments on a vision transformer?* We elected to test all the candidate methods on CIFAR-10 MLPs because they are well-known models and consistent to compare against for developing intuition about constraint methods, but do not expect such models to do well on large datasets, e.g. ImageNet-1k. For larger datasets, we elected to go forward with language models due to their importance to the field of machine learning. We think that our methods can easily be applied to vision transformers composed of linear layers, such as a ViT model.
>
> Overall, we thank you for the positive evaluation. We have updated the paper in several ways based on reviewer feedback. We added clearer explanations of the AdamW and Muon optimizers and their relevance to our analysis and expanded Appendix C with a complete and self-contained description of spectral hard/soft capping. We also improved the discussion of activation norms, $RMS \to RMS$ Lipschitz bounds, and the purpose of our MLP experiments. We believe these revisions improve both the clarity and rigor of the paper while preserving the core contributions.
>
> We look forward to discussing more with you during the discussion period, and hope to gain some clarification on the kind of experiment you are imagining.

---

### Meta-Review · Area_Chair_sSX3 · 2026-01-11

**Summary:**

The paper investigates methods for maintaining Lipschitz bounds throughout the training of Transformer models, moving beyond simple initialization constraints. Reviewers raised concerns regarding the experimental scope, with requests for Vision Transformer (ViT) on ImageNet-1K evaluations to ensure the findings generalize beyond language modeling. Due to computational constraints, the authors cannot provide such experimental results.

 Reviewers also questioned the practical utility of the methods, noting that achieving competitive accuracy often required relatively loose Lipschitz bounds, though the authors argued these bounds still offer significant improvements in activation stability and low-precision training compatibility.

Finally, some reviewers view the novelty of the proposed methods as incremental relative to existing spectral normalization / orthogonalization approaches, with unclear advantages on the Lipschitz–performance frontier.

**Reviewer Concerns:**

Addressed by Rebuttal
- Clarity of Optimizers and Methods: The authors successfully updated the text to include the formal steps for AdamW and Muon, defined the $msign$ function, and moved implementation details from the appendix to the main body.
- Empirical Frontier: The authors clarified that while Spectral Normalization is a strong baseline, their methods (specifically soft capping) provide better performance in the low-loss, low-Lipschitz regime compared to Stiefel projection.

Outstanding Concerns
- Experimental Generalization: While the authors provided reasoning for focusing on language models, the request for ImageNet-1K or Vision Transformer (ViT) results remains largely unaddressed due to compute constraints.
- Performance Gap: Reviewer QoE5 remains concerned that "competitive" performance still requires extremely loose bounds, questioning the practical utility of the $\leq 10$-Lipschitz model which shows lower accuracy.

**Reviewer Scores:**

- Reviewer rsDr: 4->4
Related work and conceptual framing improved, but concerns about novelty and experiments remain.
- Reviewer QoE5: 2->4
Many clarity and technical concerns were addressed, but doubts about the usefulness and baselines likely remain.
- Reviewer 3yVB 2->4
Explicitly stated they were raising their score to a 4 during the discussion and appreciated the clarifications on implementation and novelty.
- Reviewer WUi1 6->6:
Already positive; rebuttal did not change core view.

---

### Decision · Program_Chairs · 2026-01-26

Reject